# Neuromuscular junction instability with inactivity: morphological and functional changes after 10 days of bed rest in older adults

Evgeniia Motanova[1] , Fabio Sarto[1,2] , Samuele Negro[1], Marco Pirazzini[1,3], Ornella Rossetto[1,3,4], Michela Rigoni[1,3], Daniel W. Stashuk[5] , Mladen Gasparini[6], Boštjan Šimunic[7] , Rado Pišot[7] and Marco V. Narici[1,3]

[1]*Department of Biomedical Sciences, University of Padua, Padua, Italy*
[2]*Centre of Studies and Activities for Space "Giuseppe Colombo", University of Padua, Padua, Italy*
[3]*CIR-MYO Myology Center, University of Padua, Padua, Italy*
[4]*Institute of Neuroscience, National Research Council, Padua, Italy*
[5]*Department of Systems Design Engineering, University of Waterloo, Waterloo, ON, Canada*
[6]*Splošna Bolnišnica Izola, Izola, Slovenia*
[7]*Science and Research Center Koper, Institute for Kinesiology Research, Koper, Slovenia*

Handling Editors: Richard Carson & Jacob Thorstensen

The peer review history is available in the Supporting Information section of this article (https://doi.org/10.1113/JP288448#support-information-section).

**Abstract figure legend** This figure summarises the study of neuromuscular junction (NMJ) changes in 10 older males following a 10 day bed rest period. Baseline (BR0) and post-bed rest (BR10) assessments were conducted using

E. Motanova and F. Sarto contributed equally to this work and share first co-authorship.
Responsible for Research Governance: Professor Rosario Rizzuto, Director of the Department of Biomedical Sciences (University of Padova). Email: rosario.rizzuto@unipd.it

The Journal of Physiology

intramuscular EMG (iEMG) to measure NMJ transmission. At BR0 and BR10, muscle biopsies were obtained to analyse NMJ morphology, and blood samples were collected to evaluate C-terminal agrin fragment (CAF) levels, a biomarker of NMJ instability and acetylcholine receptor (AChR) remodelling. After the bed rest period, NMJ morphology showed reduced overlap between presynaptic and postsynaptic terminals and an increase in AChR area. Elevated CAF levels biochemically confirmed this AChR remodelling. Additionally, NMJ transmission properties were impaired, suggesting that this NMJ morphological remodelling is associated with functional alterations.

**Abstract** The neuromuscular junction (NMJ) plays a key role in modulating muscle contraction, but the impact of short-term disuse on NMJ structure and function, particularly in older humans, remains unclear. This study aimed to investigate NMJ alterations following 10 days of horizontal bed rest in 10 older males (68.5 $\pm$ 2.6 years). Before and after bed rest, vastus lateralis muscle biopsies were obtained to evaluate NMJ morphology, intramuscular EMG (iEMG) was recorded to assess NMJ function and blood samples were collected to determine circulating C-terminal agrin fragment (CAF) concentration, a biomarker of NMJ remodelling. In a sub-cohort of six participants who had NMJs in both pre- and post-bed rest biopsies, we observed altered NMJ morphology, including reduced overlap between NMJ terminals, as well as increased endplate area and perimeter. CAF concentration was elevated after bed rest, suggesting ongoing NMJ remodelling. iEMG analysis showed increased motor unit potential complexity and reduced firing rate. In addition, we observed impaired NMJ transmission, inferred from increased near-fibre jiggle and segment jitter. These findings suggest that older male individuals are susceptible to NMJ remodelling and impaired transmission with short-term disuse, providing valuable insights into the morphological and functional consequences of inactivity in an ageing population. Our study highlights the importance of developing interventions for mitigating the detrimental consequences of inactivity on neuromuscular health in older adults, which they frequently experience following injury, trauma, illness or surgery.

(Received 9 January 2025; accepted after revision 25 February 2025; first published online 15 March 2025)

**Corresponding authors** Evgeniia Motanova and Marco V. Narici: Department of Biomedical Sciences, University of Padua, Padua, 35131, Italy. Email: evgeniia.motanova@phd.unipd.it; marco.narici@unipd.it

## Key points

- The neuromuscular junction (NMJ) is crucial for signal transmission between the motoneuron and skeletal muscle, and NMJ alterations are linked to several neuromuscular disorders, as well as ageing. However, the impact of disuse on the structural and functional integrity of the NMJ, particularly in older humans, is largely unknown.
- We used the bed rest model to study the impact of inactivity on NMJ morphology and function in older men. We hypothesised that a 10 day bed rest period would lead to alterations in NMJ morphology and transmission.
- We show that 10 days of bed rest were sufficient to induce marked alterations in NMJ morphology, associated with an impaired NMJ transmission and with changes in motor unit potential properties.
- These findings suggest that older male individuals are vulnerable to NMJ dysfunction in response to inactivity and emphasise the importance of maintaining an active lifestyle for preserving neuromuscular health with ageing.

## Introduction

The neuromuscular junction (NMJ) is essential for the transmission of signals from motoneurons to skeletal muscle fibres (Rodríguez Cruz et al., 2020). Alterations at this synapse are implicated in a variety of neuromuscular disorders (Bruneteau et al., 2015; Cappello & Francolini, 2017; Navarro-Martínez et al., 2023; Verma et al., 2022) and in the context of ageing (Arnold & Clark, 2023; Deschenes & Wilson, 2003; Deschenes et al., 2022; Iyer

et al., 2021; Sarto et al., 2024). While extensive research has been conducted on NMJ morphology in rodents and small mammals, studies focusing on human NMJs, particularly in the context of ageing and muscle disuse, remain limited and controversial (Motanova et al., 2024).

It is important to note that among the few studies examining morphological alterations in human NMJs (Arizono et al., 1984; Boehm et al., 2020; Jones et al., 2017; Oda, 1984; Wokke et al., 1990), none specifically address disuse-induced changes. Previous research on NMJ alterations related to disuse has mainly relied on indirect assessments, such as biomarkers of NMJ remodelling or indicators of muscle denervation (Monti et al., 2021; Sarto et al., 2022). However, several studies involving rodent models provide valuable insights. For instance, disuse in rodents has been shown to induce NMJ fragmentation (Prakash et al., 1999), enlargement of the nerve terminal area (Fahim, 1989; Mantilla et al., 2007) and increased sprouting (Fahim, 1989). Interestingly, while younger animals do not exhibit changes in acetylcholine receptor (AChR) area during disuse, older animals have shown a reduction in AChR area (Deschenes & Wilson, 2003). In contrast, other studies have reported an expansion of the AChR area in older animals during disuse (Fagg et al., 1981; Rosenheimer, 1990; Skinner et al., 2021). Moreover, it remains unclear what happens in humans, as the effects of disuse on AChR area in human populations have not been thoroughly explored (Motanova et al., 2024). In the context of ageing, however, it has been shown that AChR area is either enlarged (Arizono et al., 1984; Oda, 1984; Wokke et al., 1990) or remains unchanged (Jones et al., 2017) with ageing. Enlargement of the AChR area can be considered detrimental as it may occur as a compensatory remodelling mechanism, independent of changes at the presynaptic terminal. In these cases, while the AChR area expands, the active zones at the presynaptic terminal, where neurotransmitter release occurs, may be reduced (Deschenes et al., 2021), or the presynaptic terminal itself may shrink (Arizono et al., 1984), leading to ineffective neurotransmission and impaired neuromuscular function despite the expansion in AChR area. These findings underline the need for further research on NMJ morphology, particularly in healthy ageing populations, as the combined effects of ageing and disuse may exacerbate NMJ vulnerability beyond what is observed with either factor alone.

The lack of data on human NMJ alterations may be partly due to technical challenges associated with isolating NMJs from muscle biopsy samples. Conventional biopsy methods yield a low success rate (∼15%) for obtaining positive NMJ samples, complicating the study of human NMJs. However, newer approaches, such as the Biopsy using Electrostimulation for Enhanced NMJ Sampling (BeeNMJ) technique, which targets regions with high NMJ concentrations, have improved success rates to 50%. In this technique, several electrodes are placed along the longitudinal axis of the muscle. A stimulator is used to deliver low-intensity pulses to the electrodes, gradually increasing until a muscle twitch is observed. The most reactive electrode is identified, and its location is marked for reproducibility. Biopsies are taken at least 2 cm away from the most reactive site along the muscle's main axis (Aubertin-Leheudre et al., 2020). Furthermore, the size of the biopsy sample is also a critical factor, with larger samples yielding higher success rates (Slater et al., 1992).

On the other hand, intramuscular EMG (iEMG), and more specifically 'near-fibre (NF) EMG', represents an excellent tool to assess NMJ function *in vivo* (Piasecki et al., 2021). However, research on the functional alterations of human NMJs with muscle disuse remains quite limited and controversial. In fact, while one study reported NMJ transmission impairment in the soleus of young to middle-aged individuals following 28 day cast immobilisation (Grana et al., 1996), no differences in NMJ transmission were found in the vastus lateralis after 10–15 days of unilateral lower limb unloading/immobilisation in young individuals (Inns et al., 2022; Sarto et al., 2022). Despite the notion that old age is often associated with periods of bed rest due to illness or trauma (Kortebein et al., 2008), the impact of inactivity on the neuromuscular health of older adults is mostly unknown.

Hence, the present study aimed to address this knowledge gap by exploring both NMJ morphological and functional changes induced by short-term bed rest in a population of older men. We hypothesised that a 10 day bed rest period would induce alterations in NMJ structure and that these would be associated with an impairment in NMJ transmission.

**Evgeniia Motanova** holds a Bachelor's degree in Cell Biology and Histology, and a Master's degree in Biology. She is currently pursuing a PhD in Biomedical Sciences at the University of Padova (Italy). Her research focuses on the alterations of neuromuscular junctions and mitochondria associated with ageing and inactivity. She is also interested in the mechanisms underlying neurodegenerative diseases. **Fabio Sarto** holds a Bachelor's and Master's degree in Sports Science, and a PhD in Biomedical Sciences. He is currently a postdoctoral researcher at the University of Padova (Italy). His research focuses on the neuromuscular alterations with inactivity, ageing and exercise, combining *in vivo* measurements with assessment of blood and muscle biomarkers.

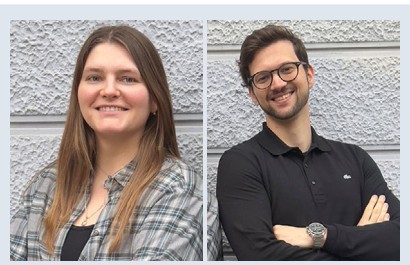

# Methods

## Ethical approval

The present study was conducted in accordance with the standards set by the latest revision of the *Declaration of Helsinki* (except for registration in a database), and was approved by the National Ethical Committee of the Slovenian Ministry of Health on 21 July 2023 with reference number 0120-123/2023/9. All participants were informed about the study's aims, procedures and potential risks prior to obtaining their written consent. After signing the written consent form, participants were enrolled in the study with the option to withdraw at any stage.

## Participants

Ten older male participants (baseline age: 68.5 ± 2.64 years; height: 172.7 ± 6.31 cm; body mass: 85.6 ± 12.25 kg) without major comorbidities or mobility impairment participated in this study. The sample size for the NMJ morphology analysis was increased by incorporating biopsies from an independent group of older adults (baseline age: 66.7 ± 2.50 years; height: 178.5 ± 4.15 cm; body mass: 86.4 ± 14.53 kg) who underwent a parallel bed rest at the same hospital for the same duration. This group followed an identical experimental design but received a multidimensional intervention, including prehabilitation, virtual reality training and protein supplementation. Dietary protein intake was increased to a total protein and amino acid intake of 1.6 ± 0.03 g/kg/day achieved by nutritional supplement (Fortifit, Nutricia, Italy), enriched with supplements of branched-chain amino acids (BCAAs) and additional free BCAAs (leucine, valine, isoleucine) (Friliver, Bracco, Italy), administered during the three main meals. Prior to the study, participants underwent a medical screening. The exclusion criteria included: regular smoking, habitual drug use, blood clotting disorders, a history of deep vein thrombosis with D-dimer levels exceeding 500 µg/l; acute or chronic skeletal, neuromuscular or cardiovascular diseases; metabolic diseases with complications; previous embolism; inflammatory diseases; psychiatric disorders; epilepsy; and the presence of ferromagnetic implants.

## Experimental protocol

The experiments were conducted at the Izola General Hospital (Izola, Slovenia), where participants underwent 10 days of horizontal bed rest. All baseline measurements (BR0) were obtained 1–2 days before the start of the bed rest period, including *in vivo* neuromuscular assessments, blood samples and muscle biopsies from the vastus lateralis. Post-bed rest measurements (BR10) included the biopsy conducted on the final day of bed rest, while all other assessments were performed on the first day of recovery. The *in vivo* tests were conducted ∼3 h after the participants were allowed to stand up. Throughout the bed rest period, participants remained in the strict lying position with only one pillow allowed for head support. Participants were prohibited from engaging in exercise-like activities or systematic voluntary movements but were allowed to switch between prone and supine positions if needed. Twenty-four-hour medical surveillance and video monitoring of the room and bathroom were conducted to ensure participants did not stand or walk. Participants remained in a horizontal position also during personal hygiene and toileting activities. Participants were provided with food four times per day (both during bed rest and the initial ambulatory period during which the baseline measurements were performed) and followed a eucaloric diet with the macronutrient distribution of 60% carbohydrates, 25% fats and 15% proteins. Participants were allowed to drink water *ad libitum*. The dietary energy requirement for each participant was calculated by multiplying their resting energy expenditure by factors of 1.2 for the bed rest period and 1.4 for the ambulatory period (Biolo et al., 2008).

## Blood collection

Blood samples were collected from the medial cubital vein at BR0 and BR10 at the same time for both time points (07.00 h), with the participants in a fasted state. Upon collection, the samples were stored at room temperature for 10 min and then centrifuged (centrifuge: CENTRIC 400, Domel, Železniki, Slovenia) at 2500 *g* (3880 r.p.m.). Following this procedure, 100 µl aliquots of plasma were prepared. The samples were subsequently stored in liquid nitrogen at −80°C until analysis.

## C-terminal agrin fragment detection

Plasma levels of C-terminal agrin fragment (CAF) were measured using commercially available enzyme-linked immunosorbent assay (ELISA) kits (Human Agrin SimpleStep ELISA®, Ab216945, Abcam, Cambridge, UK), following the manufacturer's instructions. All samples were analysed in duplicate. Plasma samples for CAF analysis were diluted 1:6 using the appropriate diluent provided with the kits. The concentrations of CAF were determined using a microplate ELISA reader (Tecan, Infinite M200, Männedorf, Switzerland). Standard curves, constructed with known and increasing concentrations of CAF, were read at 450 nm. The concentrations of CAF in the samples were then interpolated from their respective standard curves and adjusted for dilution. The coefficient of variation (CV) for the measurements was below 2%.

## Skeletal muscle tissue collection

Vastus lateralis muscle samples (∼150 mg) were obtained via biopsy using a Weil–Blakesley conchotome (Gebrüder Zepf Medizintechnik GmbH & Co. KG, Dürbheim, Germany) under local anaesthesia (2% lidocaine), ∼2 cm proximal to the central motor point: from the right leg at BR0 and the left leg at BR10. The motor point biopsy technique was chosen to increase the likelihood of detecting NMJ-positive tissue, similarly to the methodology used in the BeeNMJ study (Aubertin-Leheudre et al., 2020). The biopsy was conducted in the morning while the participants were in a fasted state, at the same time for both BR0 and BR10 (09.00 h). After the biopsy, the muscle samples were divided into several portions, with one portion (∼30 mg) designated for the analysis of NMJ morphology. This tissue was fixed for 20 min in 4% paraformaldehyde (PFA) and subsequently stored at +4°C in phosphate-buffered saline (PBS) containing 0.1% sodium azide until further tissue processing.

## Immunohistochemical staining of NMJs

After removing fat and connective tissue from muscle samples, the samples were separated into fibre bundles of 3–5 interconnected fibres. To confirm the presence of NMJs, samples were incubated overnight with $\alpha$-bungarotoxin conjugated with Alexa-555 ($\alpha$-BTx Alexa-555; 1:100; B35451, Life Technologies, Carlsbad, CA, USA), which binds to postsynaptic AChRs. Muscle fibres were then visualised using a fluorescence microscope (Leica DMi6000, Leica Microsystems, Wetzlar, Germany) to identify the presence of NMJs. Positive samples were subsequently stained using the following workflow:

(1) Quenching: incubation in 50 mM ammonium chloride ($NH_4Cl$) for 30 min, followed by PBS wash.
(2) Permeabilisation and blocking: incubation in blocking solution (PBS with 2% bovine serum albumin, 15% goat serum, 0.25% gelatin, 0.2% glycine and 0.5% Triton X-100) for 2 h, followed by PBS wash.
(3) Primary antibody incubation: samples were incubated with primary antibodies against presynaptic terminal proteins – synaptic vesicle protein 2 (SV2; 1:200), which recognises isoforms SV2A, SV2B and SV2C (SV2-s antibody was deposited with the Developmental Studies Hybridoma Bank by K. M. Buckley, Iowa, USA), and against neurofilament (Nfl) heavy polypeptide (1:1000; Ab4680, Abcam, Cambridge, UK) – for 4 days at +4°C, followed by PBS wash.

(4) Secondary antibody incubation: incubation with secondary anti-mouse antibody conjugated with Alexa Fluor 647 (1:200; A-21235, Life Technologies, Carlsbad, CA, USA), secondary anti-chicken antibody conjugated with Alexa Fluor 488 (1:200; Ab150169, Abcam, Cambridge, UK) and $\alpha$-BTx Alexa-555 (1:100; B35451, Life Technologies, Carlsbad, CA, USA).
(5) Mounting: samples were mounted using Fluorescent Mounting Medium (Dako Agilent, Santa Clara, CA, USA).

## Confocal imaging and analysis of neuromuscular junction morphology

Images were captured using a Zeiss LSM 900 confocal microscope equipped with an Airyscan 2 detector (Carl Zeiss Microscopy, Jena, Germany) and a 40×/1.4 oil immersion objective. Laser power intensity was optimised to prevent photobleaching, with excitation lines and emission ranges selected based on each fluorophore's spectral characteristics to reduce signal bleed-through across samples. Multiple z-stack series were acquired per muscle, with each stack subsequently collapsed into maximum-intensity projections and merged using ImageJ (Schneider et al., 2012), where image analyses were performed.

To quantify postsynaptic terminal morphology, the area and perimeter circumscribing postsynaptic staining were measured from maximum-intensity projections generated from 0.6 µm interval z-stacks. The analysis was conducted double-blindly, with NMJs relabelled by an independent operator before analysis, ensuring the primary operator was unaware of specific labelling. Each AChR outline was manually traced, and the enclosed surface area was measured using ImageJ. Additionally, the number of AChR clusters per NMJ was counted manually, and NMJ complexity (form factor) was calculated using the formula $(\text{perimeter}^2)/(4\pi \times \text{area})$. Nfl staining of axons was used only to distinguish between individual NMJs. Endplate occupancy was assessed using the Colocalization Threshold plugin in ImageJ (1.52v; National Institutes of Health, Bethesda, MD, USA) to quantify the overlap between SV2 (presynaptic terminal) and $\alpha$-bungarotoxin (postsynaptic terminal) staining. Endplates were classified as follows based on occupancy percentage: denervated (<40% overlap), innervated (>70% overlap) and partially denervated (40–70% overlap) (Fig. 1). This classification was inspired by similar approaches used previously for rodent NMJs (Ang et al., 2022; Comley et al., 2011; Courtney et al., 2019; Motley et al., 2011). While the studies we referenced focused on pathological conditions in genetically modified animals that created severe phenotypes, where fully denervated NMJs were defined as

those with no overlying presynaptic terminal, we opted for a milder threshold in our study to better capture less severe alterations in NMJ integrity. In our case, we detected only two NMJs with no presynaptic terminal, further justifying the use of this milder classification.

## Motor unit properties and NMJ transmission

The distal motor point of the vastus lateralis, generally located between 35% and 20% of femur length, was identified using low-intensity percutaneous electrical stimulation. Stimulations were delivered using a pen electrode connected to a Digitimer DS7AH stimulator device, applying an electrical current of 16 mA (400 V) with a pulse width of 100 µs. After locating the motor point, the current was lowered to 8–10 mA to confirm it as the most responsive site, producing the largest twitch. iEMG recordings were obtained during knee extensor contractions (90° knee angle) on an isometric dynamometer. Participants were asked to perform sub-maximal isometric contractions at 25% and 50% of maximal voluntary contraction (MVC). A 25 or 37 mm concentric needle electrode (S53155 or S53156, Teca Elite, Natus Medical Inc., Middleton, WI, USA) was employed. The needle was inserted at an angle of ∼45° at 3.6 cm

proximal to the distal motor point. Nine trapezoidal contractions were recorded: six at 25% MVC (steady-state phase of 20 s) and three at 50% MVC (steady-state phase of 10 s). The depth, angle and rotation of the needle were adjusted between contractions to sample a more heterogeneous pool of motor units. Reference electrodes were applied to the patella and patellar tendon. The iEMG signal was sampled at 40 kHz in LabChart (v.8.13, ADInstruments, Colorado Springs, CO, USA).

An experienced operator used the DQEMG software to extract motor unit potential (MUP) trains and perform all analyses (Stashuk, 1999a). The markers for MUP onset, end, positive peak and negative peak were manually adjusted, where appropriate. The inclusion criteria for MUP trains were as follows: a minimum of 34 MUPs, signal-to-noise ratios exceeding 15 a.u. and physiological waveform characteristics. Furthermore, only MUP trains with inter-discharge intervals conforming to a Gaussian distribution were included in the firing rate analysis. The following parameters were evaluated (Piasecki et al., 2021; Sarto et al., 2022; Stashuk, 1999a): (i) MUP area; (ii) MUP duration; (iii) number of turns (indicating a change in MUP waveform of at least 20 µV, reflecting MUP complexity); and (iv) mean firing rate of motor units.

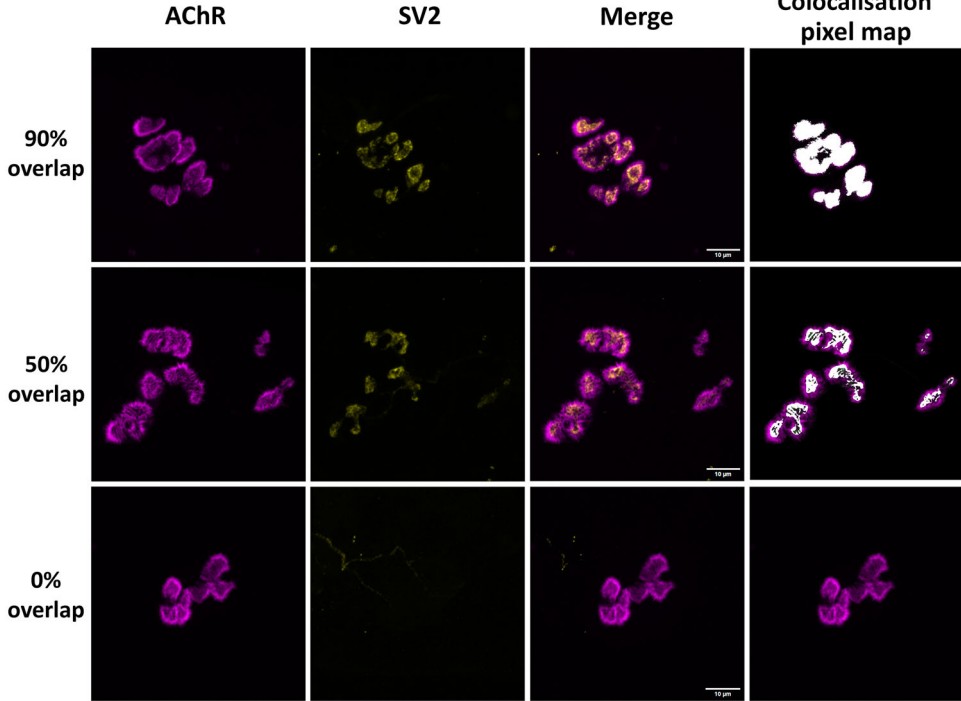

**Figure 1. Representative images of endplate occupancy assessed by colocalisation staining of pre-synaptic (synaptic vesicle protein 2; SV2) and postsynaptic (acetylcholine receptor; AChR) terminals**
Shown are representative images of fully innervated (90% overlap, row 1), partially denervated (50% overlap, row 2) and fully denervated (0% overlap, row 3) endplates. Each row shows AChR staining, SV2 staining, the merged image of both and a colocalisation pixel map generated using ImageJ's Colocalization Threshold plugin. Scale bar: 10 µm.

Near-fibre MUP (NFM) trains were obtained after applying a bandpass filter to MUPs using a second-order low-pass differentiator (Stashuk, 1999b). They were visually inspected, manually excluding any NFMs that contained activity from other motor units. Only trains containing more than 34 NFMs and an NF peak (NFPk) count greater than 1 were included. The parameters analysed for NFMs included NFM duration and NFPk count, which relates to the density of the near fibres contributing to the NFMs. NMJ transmission was assessed by NFM jiggle and NFM segment jitter, which evaluate the shape and temporal variability (mean absolute consecutive temporal differences) of consecutive NFMs, respectively. These electrophysiological parameters reflect the fluctuations in the time needed for endplate potentials to reach the threshold to generate a muscle fibre action potential and are thus considered proxies for the NMJ safety factor (Juel, 2012; Piasecki et al., 2021).

## Statistical analysis

The Shapiro–Wilk test and visual inspection of Q–Q plots were used to assess the normality of circulating biomarker concentration. Parameters that were not normally distributed (CAF concentration) were analysed using the non-parametric Wilcoxon test. Repeated measures two-way ANOVA (time × group) was used to assess the differences between groups (Old and Old + int) in response to bed rest. To account for multiple elements (NMJs or motor units) analysed per participant, NMJ morphology and iEMG parameters were evaluated using linear mixed models (Yu et al., 2022), setting time point (BR0; BR10) as a fixed effect and participant as a clustering variable. For parameters in which the model residuals were not normally distributed, generalised linear mixed models were employed, using either the gamma or inverse Gaussian distribution associated with the corresponding link function. Spearman's rank correlation coefficient was used to test associations between variables. Correlations were evaluated as small ($r \leq 0.3$), moderate ($0.30 \leq r \leq 0.70$) and large ($r \geq 0.70$). Statistical significance was set at $P \leq 0.05$. *Jamovi* software (version 2.3.1, Sydney, Australia) was used to perform linear mixed models, while all other statistical analyses were conducted using GraphPad Prism (version 8.0.1 for Windows, GraphPad Software, Boston, MA, USA). All graphs were generated using GraphPad Prism.

## Results

There were no reported side effects associated with the biopsy procedures or bed rest exposure.

## Morphological NMJ alterations accompanied by elevated CAF

At BR0, NMJ-positive samples were identified in seven participants, with the same number at BR10. Interestingly, six participants had NMJ-positive samples at both time points (BR0 and BR10). Therefore, we decided to focus the subsequent morphological analysis only on these six participants who had coincident NMJ-positive samples at both time points.

As mentioned previously, not all participants with positive NMJ samples at BR0 and BR10 originated from the same group. In parallel with our bed rest study, another group of older participants underwent bed rest exposure at the same hospital, for the same duration, and followed the same experimental design, but received a multidimensional intervention. Three of our NMJ-positive samples came from this independent group. To determine comparability between the cohorts, we compared the changes in CAF concentration and NMJ terminal overlap between the two groups and found no differences in time × group interaction in response to bed rest (CAF: $P = 0.3843$; Overlap $P = 0.9240$) (Fig. 2).

Consequently, we pooled the positive samples from both groups into a single dataset for NMJ morphological analysis. At BR0, 152 endplates were detected (mean per participant = $26 \pm 10$). At BR10, 172 endplates were detected (mean per participant = $30 \pm 12$).

Following 10 days of bed rest, a significant reduction in the overlap between presynaptic SV2 and postsynaptic AChR staining was observed ($P < 0.001$) (Fig. 3*A*, *C*, *D*). The analysis of endplate occupancy revealed a decrease in the percentage of fully innervated endplates and an increase in both partially denervated and denervated endplates after bed rest (Fig. 3*B*).

In terms of postsynaptic terminal morphology, both the AChR area ($P < 0.001$) and perimeter ($P < 0.001$) increased significantly after bed rest (Fig. 4*A*, *B*). However, there were no changes in the number of AChR fragments per NMJ (Fig. 4*C*) or the form factor (Fig. 4*D*), indicating no alterations in NMJ complexity.

To biochemically verify our findings, we assessed circulating plasma CAF concentration in the entire cohort, as it is considered a biomarker associated with NMJ remodelling (Drey et al., 2013; Monti et al., 2021). Our analysis revealed a significant 9% increase in CAF concentration following 10 days of bed rest ($P = 0.019$) (Fig. 5).

## Changes in motor unit properties and NMJ transmission

In light of these NMJ morphological and biochemical alterations, we employed iEMG to test their potential electrophysiological impact at the motor unit level. Two

participants were excluded from the analysis at 50% MVC due to the absence of suitable MUPs trains. Regarding MUP analysis, data from 618 (mean per participant: 30.9 ± 11.2) and 215 MUs (mean per participant: 13.4 ± 7.7), sampled at 25% and 50% MVC, respectively, were included. Regarding NFM analysis, data from 352 and 107 MUs, sampled at 25% (mean per participant: 17.6 ± 7.2) and 50% (mean per participant: 6.7 ± 5.1) MVC, respectively, were included.

Following bed rest, mean firing rate was significantly decreased at both contraction intensities (25% MVC: $P < 0.001$; 50% MVC: $P = 0.0195$) (Fig. 6*D*). Additionally, iEMG analysis revealed an increased number of turns only at 25% MVC ($P = 0.0059$) (Fig. 6*C*), suggesting an elevated MUP complexity induced by disuse. MUP area declined only at 25% MVC ($P = 0.0016$) (Fig. 6*A*) while showing a tendency to increase for MUP duration only at 50% MVC ($P = 0.051$) (Fig. 6*B*). Regarding NFM parameters (Fig. 7), NFPk count was increased at both contraction intensities (25% MVC: $P < 0.001$; 50% MVC: $P = 0.0092$), while NFM duration was increased at 25% only ($P < 0.001$). Finally, our findings revealed an impairment in NMJ transmission, as suggested by increased NFM jiggle ($P < 0.001$) and NFM segment jitter ($P = 0.003$) at 25% MVC, but not at 50% MVC.

### Correlation analysis between NMJ morphology and NMJ transmission properties

Given the observed NMJ morphological remodelling, we hypothesised that changes in the overlap between the presynaptic and postsynaptic terminals may be associated with functional impairments in NMJ transmission and motor unit properties. Additionally, since increased circulating CAF concentration has been considered a biomarker for NMJ remodelling (Monti et al., 2023; Pratt et al., 2024; Sarto et al., 2024), we were interested in exploring whether morphological changes in the NMJ could be linked to changes in CAF concentration in the blood. To investigate these potential interactions, we conducted a series of correlation analyses examining the relationships between NMJ terminal overlap, postsynaptic AChR terminal morphology, electrophysiological parameters of motor units and CAF concentration. For this analysis, we decided to employ only iEMG parameters at 25% MVC, considering the larger dataset of motor units obtained and the more consistent findings with disuse.

We first examined whether changes in the overlap between the presynaptic and postsynaptic NMJ terminals correlated with both biochemical (CAF concentration) and electrophysiological parameters. The results of this analysis are reported in Table 1. Of note, we were unable to detect any significant correlations between the above-mentioned parameters. However, we identified strong correlations between MUP mean firing rate and overlap ($r = -0.80$; $P = 0.133$), MUP duration and overlap ($r = -0.90$; $P = 0.083$), and MUP number of turns and overlap ($r = -0.80$; $P = 0.133$), suggesting a potential link between motor unit electrophysiological properties and NMJ morphology.

Next, we explored whether changes in the morphology of postsynaptic AChR terminals were associated with circulating CAF concentration and NMJ electrophysiological measures, as reported in Table 2. Again, we found some strong but non-significant correlations

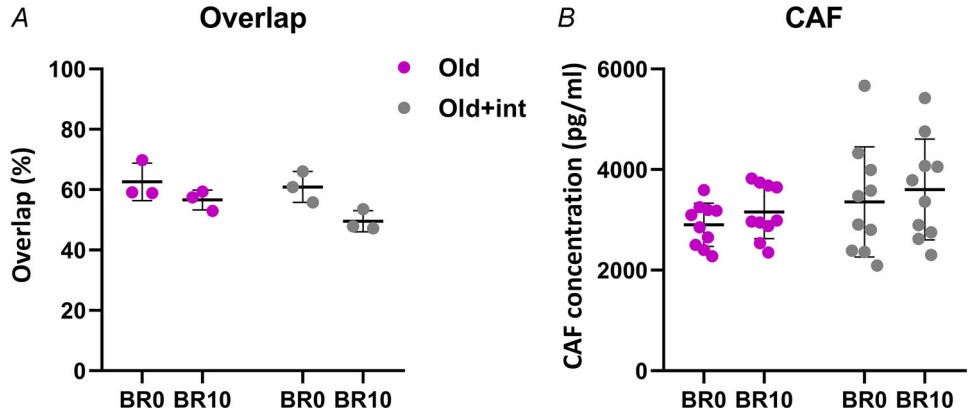

**Figure 2. Comparison of bed rest effects on the neuromuscular junction (NMJ) terminals overlap and C-terminal agrin fragment (CAF) levels in older participants with and without interventions**
*A*, change in the overlap between presynaptic and postsynaptic terminals in older participants (Old) and older participants with interventions (Old + int), measured between BR0 and BR10 (before and after 10 days of bed rest); *N* = 3 (each group). Statistical analysis was performed using repeated-measures two-way ANOVA. *B*, changes in CAF concentration in older participants (Old) and older participants with interventions (Old + int), measured between BR0 and BR10; *N* = 10 (each group). Statistical analysis was performed using repeated-measures two-way ANOVA.

between NFM segment jitter and AChR area ($r = -0.70$; $P = 0.233$), and NFM jiggle and AChR area ($r = 0.90$; $P = 0.083$), outlining a potential link between NMJ transmission properties and morphological integrity of NMJs. Interestingly, we found a strong and significant correlation between the number of AChR fragments per NMJ and CAF concentration in blood ($r = -0.94$; $P = 0.017$), which is known to be released in circulation upon cleavage of AChR-clustering protein agrin, resulting in AChR dispersion (Monti et al., 2023). However, all these correlations have to be interpreted with extreme caution, as they have been performed on just six participants, which is a very limited sample size for this type of analysis to consider results robust.

## Discussion

In this study, we investigated morphological, biochemical and functional changes of the human NMJ in response to a 10 day bed rest intervention in older male individuals. For the first time, we report changes in human NMJ morphology and function induced by inactivity. Our findings demonstrate significant morphological changes in NMJs, including reduced terminal overlap and alterations in AChR area and perimeter at postsynaptic terminals, together with elevated circulating CAF levels, impaired NMJ transmission and altered motor unit properties following inactivity.

### NMJ morphological and biochemical alterations

While NMJ remodelling with disuse has been well documented in animal models (Deschenes & Wilson, 2003; Deschenes et al., 2021; Fahim, 1989; Mantilla et al., 2007; Prakash et al., 1999), there has been a significant gap in understanding how disuse affects the morphology of human NMJs due to the methodological challenges of detecting positive NMJ tissue.

One of the most striking findings of our study was the reduction in the overlap between presynaptic and post-synaptic terminals following bed rest. The decreased overlap at BR10 suggests a shift toward partial denervation or reduced innervation at the NMJ, as evidenced by the increased proportion of partially denervated and denervated endplates. These findings build on previous studies that inferred altered innervation profile and NMJ instability with disuse indirectly through the assessment of circulating and muscle biomarkers (Arentson-Lantz et al.,

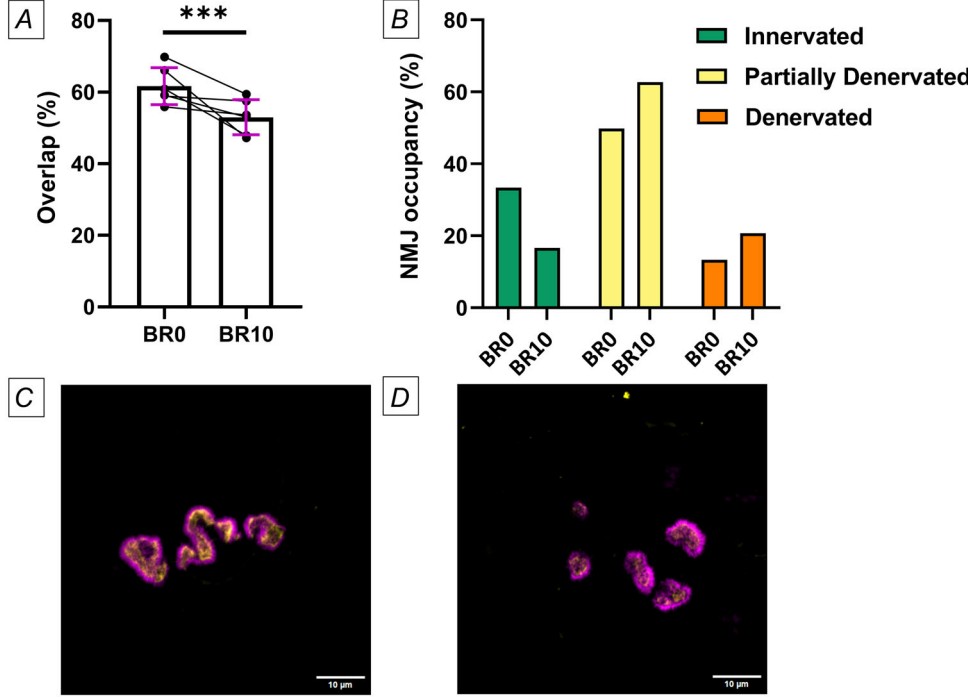

**Figure 3. Alterations of the neuromuscular junction (NMJ) terminals overlap and NMJ occupancy**
*A*, overlap between the NMJ presynaptic and postsynaptic terminals assessed before bed rest (BR0) and after 10 days of bed rest (BR10). *B*, percentage distribution of different NMJ categories (innervated, partially denervated and fully denervated) before bed rest (BR0) and after bed rest (BR10). *C* and *D*, representative NMJs from the same participant, with the NMJ before bed rest (BR0) and an overlap of 82% (*C*), and the NMJ after bed rest (BR10) with an overlap of 42% (*D*). BR0 – 152 NMJs detected (mean per participant = 26 ± 10). BR10 – 172 NMJs (mean per participant = 30 ± 12). Statistical analysis was performed using linear mixed models. Results are shown as mean and standard deviation. Scale bar: 10 μm. *N* = 6. ***$P \leq 0.001$.

2016; Demangel et al., 2017; Monti et al., 2021; Sirago et al., 2023). While no previous human studies have specifically assessed the overlap between presynaptic and postsynaptic terminals in the context of disuse, one investigation found no changes in this parameter comparing young and old individuals (Jones et al., 2017). In this study, however, the tissue was harvested following lower limb amputations, which may have influenced the results due to underlying patient pathologies. Unfortunately, the analysis of NMJ terminal overlap and their categorisation into denervated, partially denervated and fully innervated states was conducted on a limited cohort of participants. As mentioned earlier, this limitation stems from the technical challenges of identifying NMJs in human biopsies. We believe that future research involving a larger cohort of older adults, stratified by age intervals and accounting for physical activity levels, is necessary to thoroughly investigate the changes in NMJs associated with inactivity in ageing populations. While it is interesting that human NMJs appear stable across the lifespan (Jones et al., 2017), our study suggests they are affected by periods of inactivity. We also suggest that similar research be extended to other age groups (young and middle-aged), with stratification based on physical activity levels.

In addition to the reduced terminal overlap, we observed significant changes in the morphology of the postsynaptic AChR terminal, including increased area and perimeter. These changes, probably a response to an ongoing denervation process, are consistent with findings on disuse in older animals (Deschenes & Wilson, 2003). Interestingly, in this previous study, the authors also found no changes in AChR area in young animals who underwent the same period of disuse as older animals, suggesting NMJ stability in younger animals during prolonged inactivity. In contrast, shorter periods of disuse in young mice resulted in a reduction in endplate size (Chibalin et al., 2018; Rocha et al., 2020). This suggests that younger animals may be more susceptible to NMJ damage during the early stages of disuse but may be able to restore NMJ integrity over longer periods, whereas older animals may lack this adaptive capacity. However, in these studies, NMJ transmission properties were not assessed, making it unclear if the NMJs were functionally stable. To date, the causal relationship between morphological changes and functional impairment remains controversial – whether structural alterations precede or follow functional changes. For instance, ageing has been associated with reduced synaptic area and AChR fragmentation,

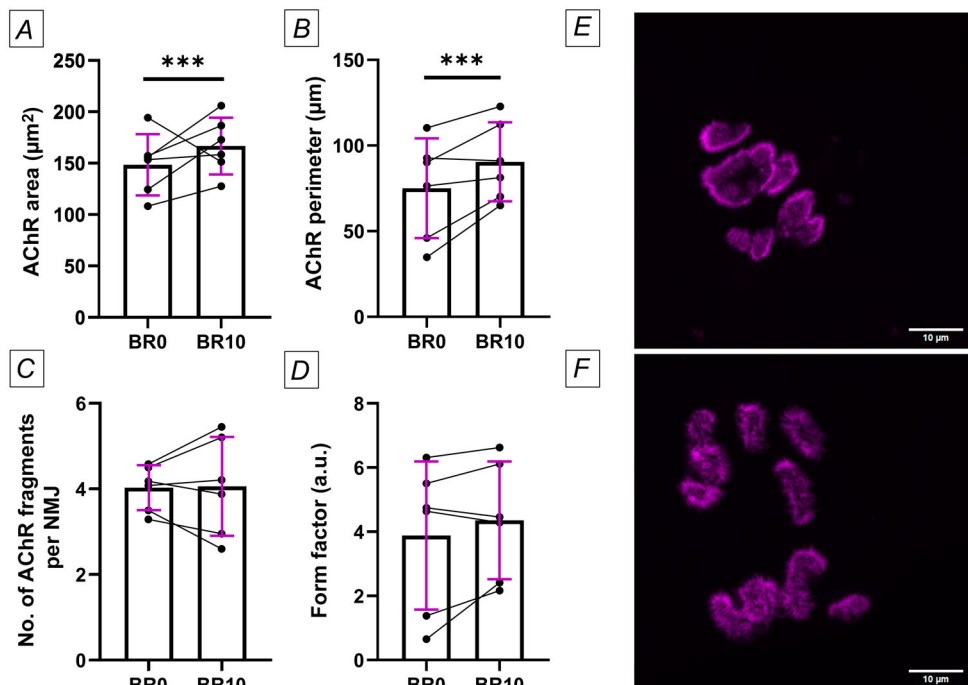

**Figure 4. Morphological alterations of the neuromuscular junction postsynaptic terminal (acetylcholine receptors; AChR)**
*A*, AChR area; *B*, AChR perimeter; *C*, the number of AChR fragments per NMJ; and *D*, form factor, all assessed before bed rest (BR0) and after 10 days of bed rest (BR10). *E* and *F*, representative images of AChR staining before bed rest (BR0) (*E*) and after 10 days of bed rest (BR10) (*F*). BR0 – 152 NMJs detected (mean per participant = 26 ± 10). BR10 – 172 NMJs (mean per participant = 30 ± 12). Statistical analysis was performed using linear mixed models. Results are shown as mean and standard deviation. Scale bar: 10 µm. $N = 6$. ***$P \leq 0.001$.

**Table 1. Correlations between neuromuscular junction (NMJ) terminals overlap, C-terminal agrin fragment (CAF) concentration and electrophysiological parameters (percentage changes before and after bed rest)**

| | CAF concentration | NFM segment jitter at 25% MVC | NFM jiggle at 25% MVC | MUP mean firing rate at 25% MVC | MUP area at 25% MVC | MUP duration at 25% MVC | MUP number of turns at 25% MVC | NFPk count at 25% MVC | NFM duration at 25% MVC |
|---|---|---|---|---|---|---|---|---|---|
| Overlap | r = 0.25<br>P = 0.658 | r = 0.20<br>P = 0.783 | r = 0.10<br>P = 0.950 | r = −0.80<br>P = 0.133 | r = −0.30<br>P = 0.683 | r = −0.90<br>P = 0.083 | r = −0.80<br>P = 0.133 | r = 0.00<br>P = 1.000 | r = −0.30<br>P = 0.683 |

Abbreviations: MVC, maximum voluntary contraction; NFM, near-fibre motor unit potential.

**Table 2. Correlations between postsynaptic terminal morphology, C-terminal agrin fragment (CAF) concentration and electrophysiological parameters (percentage changes before and after bed rest)**

| | CAF concentration | NFM segment jitter at 25% MVC | NFM jiggle at 25% MVC | MUP mean firing rate at 25% MVC | MUP area at 25% MVC | MUP duration at 25% MVC | MUP number of turns at 25% MVC | NFM count at 25% MVC | NFM duration at 25% MVC |
|---|---|---|---|---|---|---|---|---|---|
| AChR area | r = −0.54<br>P = 0.297 | r = −0.70<br>P = 0.233 | r = 0.90<br>P = 0.083 | r = −0.20<br>P = 0.783 | r = −0.70<br>P = 0.23 | r = −0.10<br>P = 0.950 | r = 0.30<br>P = 0.683 | r = −0.50<br>P = 0.450 | r = 0.30<br>P = 0.683 |
| AChR perimeter | r = −0.43<br>P = 0.419 | r = −0.60<br>P = 0.350 | r = 0.70<br>P = 0.233 | r = 0.10<br>P = 0.950 | r = −0.60<br>P = 0.35 | r = 0.30<br>P = 0.683 | r = 0.60<br>P = 0.350 | r = −0.60<br>P = 0.350 | r = 0.50<br>P = 0.450 |
| Form factor | r = −0.48<br>P = 0.356 | r = −0.50<br>P = 0.450 | r = 0.60<br>P = 0.350 | r = 0.20<br>P = 0.783 | r = −0.50<br>P = 0.45 | r = 0.10<br>P = 0.950 | r = 0.30<br>P = 0.683 | r = −0.70<br>P = 0.233 | r = 0.30<br>P = 0.683 |
| No. of AChR fragments per NMJ | r = −0.94<br>P = 0.017 | r = −0.60<br>P = 0.350 | r = 0.50<br>P = 0.450 | r = 0.60<br>P = 0.350 | r = −0.10<br>P = 0.95 | r = 0.30<br>P = 0.683 | r = 0.40<br>P = 0.517 | r = −0.40<br>P = 0.517 | r = 0.10<br>P = 0.950 |

Abbreviations: MVC, maximum voluntary contraction; NFM: near-fibre motor unit potential.

though these changes did not correlate with transmission deficits (Willadt et al., 2016). Conversely, another study found no morphological changes in the postsynaptic terminals despite reductions in active zone staining and pre- to postsynaptic coupling (Deschenes et al., 2021). Although this reduction suggests potential functional alterations via decreased synaptic vesicle trafficking and impairment of neuromuscular transmission, the actual transmission properties were not assessed, and thus active zone remodelling does not necessarily imply impaired signal transmission. Notably, we found no changes in the number of AChR fragments or the form factor, indicating that the overall complexity of the NMJ remained unchanged.

To support the morphological changes observed at the NMJ, we performed a biochemical analysis of circulating CAF concentration. CAF, a fragment of agrin – a protein responsible for the clustering of AChRs – is released into circulation following cleavage by neurotrypsin, and this process is directly related to the remodelling of the NMJ (Bolliger et al., 2010; Stephan et al., 2008). Our results showed a significant increase in CAF concentration after 10 days of bed rest, confirming the presence of NMJ remodelling, in line with previous observations of our group in young individuals exposed to short-term disuse (Monti et al., 2021; Sarto et al., 2022). The strong negative correlation between circulating CAF levels and the number of AChR fragments per NMJ appears to support the idea that CAF levels may be linked to the

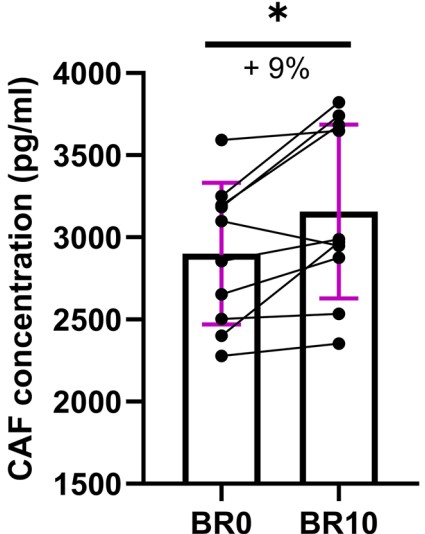

**Figure 5. Circulating C-terminal agrin fragment (CAF) concentration before (BR0) and after 10 days (BR10) of bed rest**
Individual values of CAF concentration for each participant are shown at BR0 and BR10 time points. Statistical analysis was performed using a Wilcoxon test. Results are shown as mean and standard deviation. $N = 10$. *$P = 0.019$.

remodelling of the postsynaptic AChR area, a hypothesis that is consistent with the notion that the agrin receptor is selectively concentrated in postsynaptic membranes (Bolliger et al., 2010; Stephan et al., 2008).

In essence, the observed reduction in the overlap of NMJ terminals and changes in postsynaptic terminal morphology (increased area and perimeter), along with the biochemical evidence of increased CAF levels, strongly suggest that inactivity induces significant structural changes at the NMJ in older male individuals.

## Alterations in motor unit properties and NMJ transmission

In addition to these unique morphological analyses, this is also the first investigation assessing the impact of muscle disuse on motor unit electrophysiological properties in older male individuals.

We observed significant decreases in firing rate – the rate at which motoneurons transmit signals to muscle fibres, playing a critical role in generating and maintaining muscle force – following the 10 day bed rest period. This is in agreement with previous literature in younger populations across different disuse models and muscle groups (Duchateau & Hainaut, 1990; Inns et al., 2022; Sarto et al., 2022; Seki et al., 2001, 2007; Valli et al., 2024). An interesting observation is that in our study firing rate was decreased at both low (25% MVC) and moderate (50% MVC) intensities, differently from what was previously reported during a disuse period of the same duration in young individuals (Valli et al., 2024). While a direct comparison with this previous study is complex due to the different disuse models (bed rest *vs.* unilateral lower limb suspension) and EMG techniques employed (iEMG *vs.* high-density surface EMG), it is tempting to speculate that the lower firing rates at moderate contraction intensities may represent a distinct characteristic of older populations in response to short-term disuse, possibly due to an impairment of higher-threshold motor units. This suppression in firing rate may be due to axonal structural damage, observed with disuse in some animal model studies (Canu et al., 2009; de-Doncker et al., 2006) and/or decreased neuromodulatory contribution, inferred previously in young individuals with short-term disuse in a previous investigation from our group (Martino et al., 2024).

In line with a recent 14 day cast immobilisation study (Inns et al., 2022), we also noticed a reduced MUP area, reflecting a smaller motor unit size, probably due to muscle fibre atrophy following bed rest, as previously reported in other bed rest studies in older individuals (Dirks et al., 2016; Moore et al., 2018; Standley et al., 2020). In addition, our results showed robust changes in MUP turns, NFM duration and NFPk count, overall

highlighting increased temporal dispersion of propagating muscle fibre action potentials at the recording point. A potential mechanism underlying these differences in conduction times along axonal branches and/or muscle fibres could be initial denervation/reinnervation processes (Piasecki et al., 2021). This hypothesis is supported by our study's direct morphological assessment, which revealed a decrease in the proportion of fully innervated NMJs in favour of an increase in partially and fully denervated NMJs (see above). The longitudinal associations observed between terminal overlap and some of these electrophysiological parameters appear to support this concept (Table 1).

Finally, we observed an increased NFM jiggle and NFM segment jitter, suggesting that NMJ function was also altered following the bed rest period. As previous studies in young individuals reported no changes in these parameters with disuse of short disuse durations (10–15 days) using a similar iEMG set-up (Inns et al., 2022; Sarto et al., 2022), our findings suggest that older male individuals may be more prone to NMJ transmission impairment in response to short-term disuse compared to young populations, an observation that appears clinically relevant. Additionally, while relationships between NMJ structure and function remain poorly understood in both animal models and humans (Slater, 2020), our study supports the view that NMJ morphology and function may be connected, as evidenced by correlations between NMJ terminal overlap, postsynaptic AChR morphology and NMJ electrophysiological parameters (Tables 1 and 2). However, as mentioned previously, the results of the correlation analysis should be interpreted with extreme caution, as they were performed on a very limited sample size that could lead to spurious associations.

Overall, by focusing on older male participants, our study aimed to extend the current understanding of the impact of inactivity on the neuromuscular system of older individuals. Our findings may be particularly relevant given the increasing prevalence of conditions that lead to periods of immobility in older adults, such as hospitalisation or highly sedentary lifestyles (Harvey et al., 2013). Notably, most older adults already show a decline in these neuromuscular features compared to younger populations (Connelly et al., 1999; Piasecki et al., 2016; Sarto et al., 2024), making them particularly vulnerable to

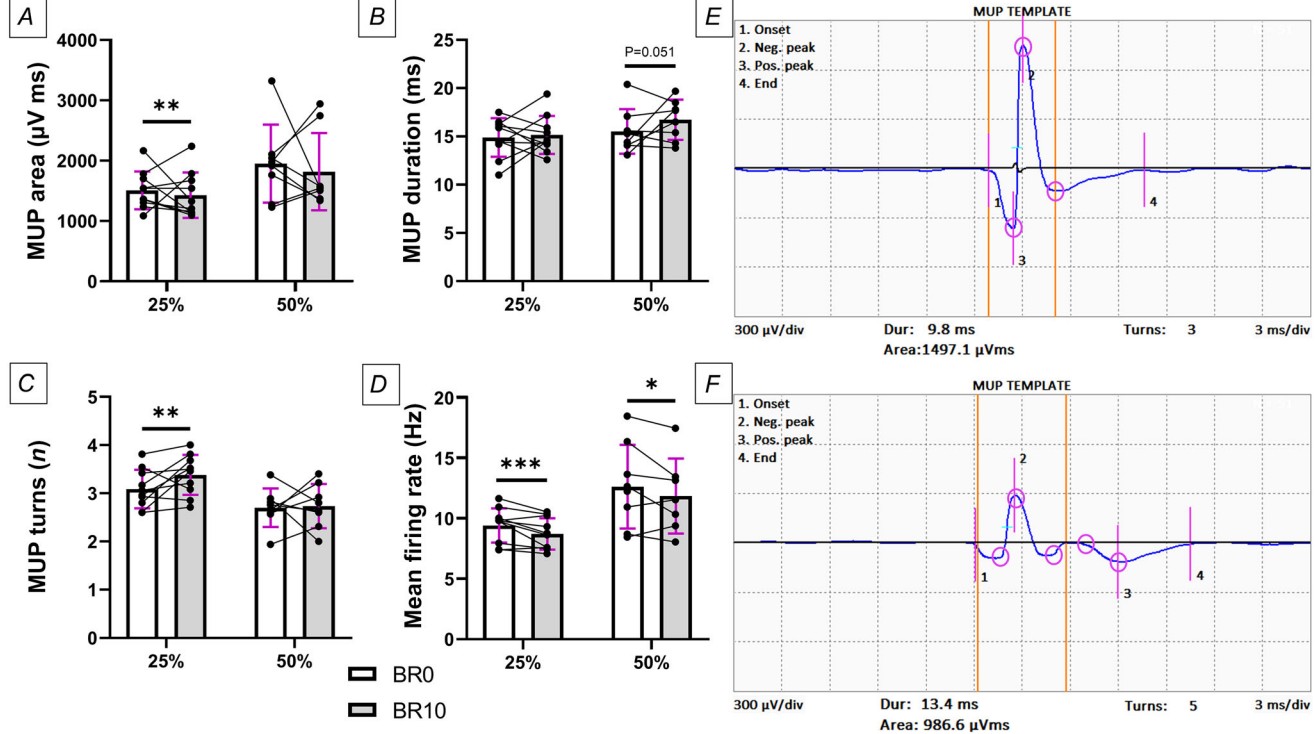

**Figure 6. Changes in motor unit potentials (MUP) recorded using intramuscular EMG**
*A*, MUP area; *B*, MUP duration; *C*, MUP turns; and *D*, mean firing rate, all assessed before bed rest (BR0) and after 10 days of bed rest (BR10). *E* and *F*, representative MUP templates at 25% of maximum voluntary contraction before bed rest (BR0) (*E*) and after 10 days of bed rest (BR10) (*F*) recorded from the same participant. MUP area represents the area under the MUP waveform. MUP duration was calculated as the time between markers 1 and 4, with each turn indicated by a magenta circle. Statistical analysis was performed using generalised linear mixed models. Results are shown as mean and standard deviation. *N* = 10 at 25% MVC; *N* = 8 at 50% MVC. *$P \leq 0.05$; **$P \leq 0.01$; ***$P \leq 0.001$.

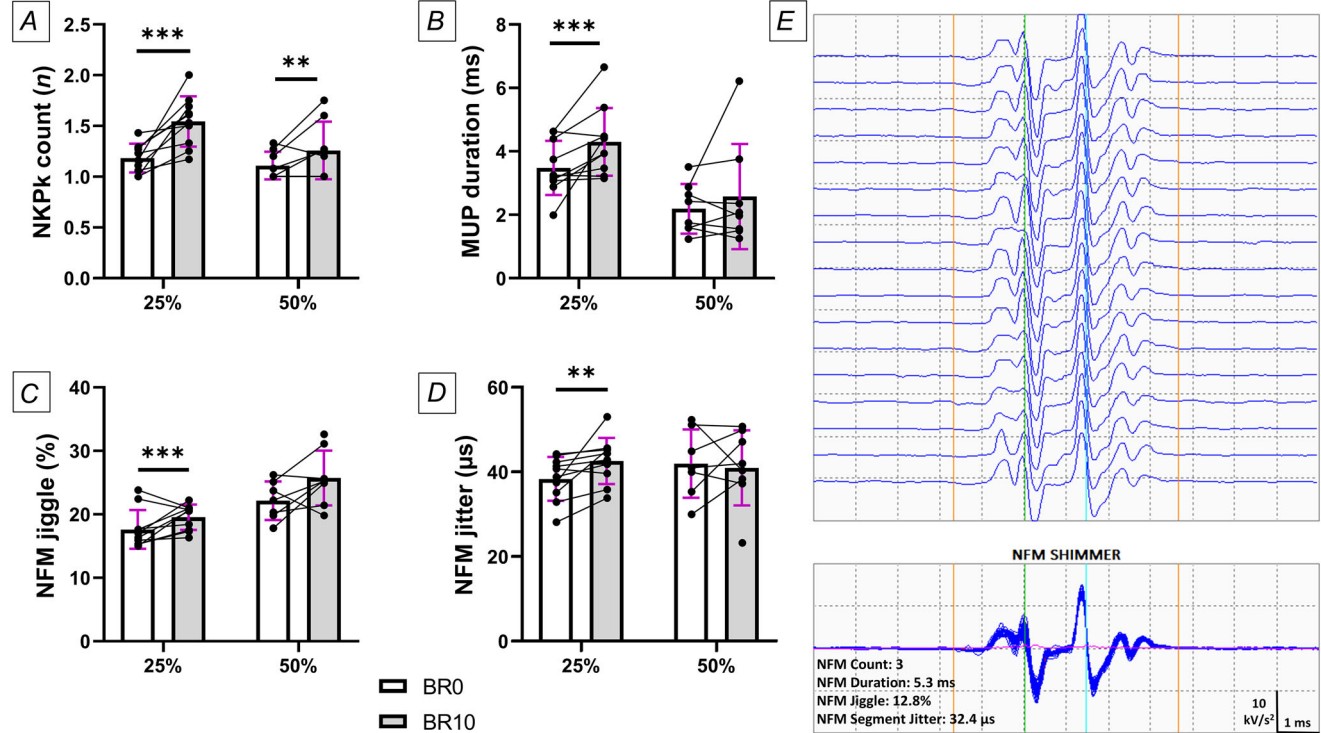

**Figure 7. Changes in near-fibre EMG parameters obtained using intramuscular EMG**
*A*, near-fibre peak (NFPk) count; *B*, near-fibre motor unit potential (NFM) duration; *C*, NFM jiggle; and *D*, NFM segment jitter, all assessed before bed rest (BR0) and after 10 days of bed rest (BR10). *E*, representative NFM raster and shimmer plot at 25% of maximum voluntary contraction. Statistical analysis was performed using generalised linear mixed models. Results are shown as mean and standard deviation. *N* = 10 at 25% MVC; *N* = 8 at 50% MVC. **$P \leq 0.01$; ***$P \leq 0.001$.

further deterioration in response to periods of inactivity or immobilisation due to illness or trauma.

## Limitations and future directions

We acknowledge that this study is based on a small sample size, with NMJ analysis performed on only six participants. This limitation arises from the technical challenges associated with the difficulty of obtaining muscle samples containing NMJs in humans. Additionally, our findings are limited to male individuals, as older females were not included due to their increased susceptibility to disuse muscle atrophy and more profound functional alterations in atrophying conditions (Della Peruta et al., 2023), as well as the higher risk of frailty, which could be exacerbated by prolonged bed rest (Gordon et al., 2017). Additionally, immobilisation in older females is associated with a specific pattern of cortical bone loss and reduction in osteocyte density that exceeds the severity observed in postmenopausal osteoporosis, raising ethical concerns about exposing them to excessive risk (Rolvien et al., 2020). We recognise that exploring potential sex-related differences should be a priority of future studies.

## Conclusion

A 10 day bed rest intervention performed in older male individuals resulted in changes in NMJ morphology, including reduced presynaptic and postsynaptic terminal overlap, alterations in postsynaptic AChR morphology and elevated circulating CAF levels, all indicative of significant NMJ remodelling. Additionally, impairments in motor unit properties and NMJ transmission were observed, suggesting that older individuals may be particularly vulnerable to inactivity-induced NMJ dysfunction. These novel findings emphasise the importance of understanding how disuse affects neuromuscular health in ageing populations. Future studies should aim to identify the underlying mechanisms driving NMJ remodelling and explore potential interventions to prevent or reverse the functional impairments associated with disuse in older adults.

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

## Additional information

### Data availability statement

The collection of *z*-stacks acquired in this study is openly available and can be accessed via the following link: https://osf.io/6hcyb/. This dataset includes all relevant images and data used in the analysis of NMJ morphology presented in this paper. The other data that support the findings of this study will be made available from the corresponding author upon reasonable request.

### Competing interests

None declared.

## Author contributions

Conceptualisation, M.V.N., R.P., B.S.; Methodology, M.V.N., B.S., E.M., F.S., S.N., M.P., D.S.; Data analysis, E.M., F.S.; Writing – original draft, E.M. and F.S.; Writing – review and editing, M.V.N., E.M., F.S., S.N., D.S., M.P., O.R., M.R., R.P., B.S. Project administration, M.V.N., B.S., R.P.; Funding acquisition, M.V.N., E.M.; Resources, M.V.N., B.S., R.P.; Supervision, M.V.N.

All authors reviewed, revised, and approved the final version of the manuscript. They also agreed to take full responsibility for all aspects of the work, ensuring that any questions regarding its accuracy or integrity are thoroughly investigated and addressed.

## Funding

The present study was funded by the PRIN project ('InactivAge' n. 2020EM9A8X) to M.V.N. We also acknowledge co-funding from Next Generation EU to M.V.N. in the context of the National Recovery and Resilience Plan, Investment PE8 – Project Age-It: 'Ageing Well in an Ageing Society'. This work has received funding from the European Union's Horizon 2020 research and innovation programme under the Marie Skłodowska-Curie grant agreement 101034319 and from the European Union – NextGeneration EU. This study was co-funded by the ARIS ('J5-4593'), while the parallel bed rest study, involving participants undergoing the multimodal intervention, was co-financed by the European Union under the Interreg VI-A Italy-Slovenia Programme ('X-BRAIN.net' project). The views and opinions expressed are only those of the authors and do not necessarily reflect those of the European Union or the European Commission. Neither the European Union nor the European Commission can be held responsible for them.

## Acknowledgements

The authors extend their sincere gratitude to the participants for their time and effort in contributing to the data collection. Special thanks to Dr Miloš Kalc for help during iEMG data collection, the medical staff at Izola Hospital for their dedicated care of the participants, and the staff of the Koper Science and Research Centre (Institute for Kinesiology Research) for their crucial role in organising and supporting the participants throughout the study.

Open access publishing facilitated by Universita degli Studi di Padova, as part of the Wiley - CRUI-CARE agreement.

## Keywords

ageing, C-terminal agrin fragment, disuse, electromyography, motor unit, NMJ, unloading

## Supporting information

Additional supporting information can be found online in the Supporting Information section at the end of the HTML view of the article. Supporting information files available:

**Peer Review History**

