## [Peer Review History · The Journal of Physiology]

Neuromuscular Junction Instability with Inactivity: Morphological and Functional Changes After 10-Day Bed Rest in Older Adults

Evgeniia Motanova, Fabio Sarto, Samuele Negro, Marco Pirazzini, Ornella Rossetto, Michela Rigoni, Dan Stashuk, Mladen Gasparini, Bostjan Simunic, Rado Pisot, and Marco Narici

DOI: 10.1113/JP288448

Corresponding author(s): Evgeniia Motanova (evgeniia.motanova@phd.unipd.it)

The following individual(s) involved in review of this submission have agreed to reveal their identity: Jason M DeFreitas (Referee #1)

Review Timeline:

Submission Date:	09-Jan-2025
Editorial Decision:	05-Feb-2025
Revision Received:	17-Feb-2025
Accepted:	25-Feb-2025

Senior Editor: Richard Carson

Reviewing Editor: Jacob Thorstensen

Transaction Report:

Dear Dr Motanova,

Re: JP-RP-2025-288448 "Neuromuscular Junction Instability with Inactivity: Morphological and Functional Changes After 10-Day Bed Rest in Older Adults" by Evgeniia Motanova, Fabio Sarto, Samuele Negro, Marco Pirazzini, Ornella Rossetto, Michela Rigoni, Dan Stashuk, Mladen Gasparini, Bostjan Simunic, Rado Pisot, and Marco Narici

Thank you for submitting your manuscript to The Journal of Physiology. It has been assessed by a Reviewing Editor and by 2 expert referees and we are pleased to tell you that it is potentially acceptable for publication following satisfactory major revision.

Please address all the points raised and incorporate all requested revisions or explain in your Response to Referees why a change has not been made. We hope you will find the comments helpful and that you will be able to return your revised manuscript within 2 months. If you require longer than this, please contact journal staff: jp@physoc.org. Please note that this letter does not constitute a guarantee for acceptance of your revised manuscript.

LANGUAGE EDITING AND SUPPORT FOR PUBLICATION: If you would like help with English language editing, or other article preparation support, Wiley Editing Services offers expert help, including English Language Editing, as well as translation, manuscript formatting, and figure formatting at www.wileyauthors.com/eoo/preparation. You can also find resources for Preparing Your Article for general guidance about writing and preparing your manuscript at www.wileyauthors.com/eoo/prepresources.

REVISION CHECKLIST:

We look forward to receiving your revised submission.

Yours sincerely,

Richard Carson
Senior Editor
The Journal of Physiology

REQUIRED ITEMS

- Author photo and profile. First or joint first authors are asked to provide a short biography (no more than 100 words for one author or 150 words in total for joint first authors) and a portrait photograph. These should be uploaded and clearly labelled together in a Word document with the revised version of the manuscript. See Information for Authors for further details.

- You must start the Methods section with a paragraph headed Ethical Approval. If experiments were conducted on humans, confirmation that informed consent was obtained, preferably in writing, that the studies conformed to the standards set by the latest revision of the Declaration of Helsinki and that the procedures were approved by a properly constituted ethics committee, which should be named, must be included in the article file. If the research study was registered (clause 35 of the Declaration of Helsinki), the registration database should be indicated, otherwise the lack of registration should be noted as an exception (e.g. The study conformed to the standards set by the Declaration of Helsinki, except for registration in a database). For further information see: <https://physoc.onlinelibrary.wiley.com/hub/human-experiments>.

- Please include an Abstract Figure file, as well as the Figure Legend text within the main article file. The Abstract Figure is a piece of artwork designed to give readers an immediate understanding of the research and should summarise the main conclusions. If possible, the image should be easily 'readable' from left to right or top to bottom. It should show the physiological relevance of the manuscript so readers can assess the importance and content of its findings. Abstract Figures should not merely recapitulate other figures in the manuscript. Please try to keep the diagram as simple as possible and without superfluous information that may distract from the main conclusion(s). Abstract Figures must be provided by authors no later than the revised manuscript stage and should be uploaded as a separate file during online submission labelled as File Type 'Abstract Figure'. Please also ensure that you include the figure legend in the main article file. All Abstract Figures should be created using BioRender. Authors should use The Journal's premium BioRender account to export high-resolution images. Details on how to use and access the premium account are included as part of this email.

EDITOR COMMENTS

Reviewing Editor:

This is a comprehensive study examining the effects of physical inactivity on neuromuscular junction structure and function in older individuals. It has been reviewed by two experts in neuromuscular physiology, particularly in the context of some of the experimental approaches you used (i.e., EMG and neuromuscular junction morphology assessed through biopsies). Both reviewers were generally favourable of your submission and have commented of the novelty of the work. Regardless, they raised some issues, and have made several constructive comments that need to be addressed. These issues primarily relate to the low sample size, and the fact that only male subjects were recruited. This affects how generalisable the findings are and need to be discussed in more detail to caution the over extrapolation of findings. There were also some issues raised regarding the methodology and statistical approach, which are likely addressed by re-analysing data/samples. I hope this feedback serves useful. As an additional observation, please note that error bars are not defined in Figure 2.

Please also see 'Required Items' above.

Senior Editor:

As noted by the referees and the Reviewing Editor, this work has many strengths. This being the case, it is recommended that the authors omit from their presentation, analyses that (due to the small sample size) are unlikely to appear convincing to readers. For example, and as highlighted in the reviews, the correlations have very low statistical power and, given the large number undertaken, are very likely to have yielded spurious false positive outcomes. More generally, the limitations on interpretation imposed by the small sample size should be emphasised.

REFeree COMMENTS

Referee #1:

This study examines NMJ morphology and function after 10 days of bed rest in older adult males. The authors are to be commended for the ambitious nature of this study. While the sample size is a bit low, the study design itself is very ambitious and provides incredibly valuable information regarding changes to our NMJs during disuse. My comments and concerns are listed below:

- I do not understand why only males are included in this study. The only mention is at the very end, which it is stated that older women have a higher risk of frailty. I do think this needs to be justified and more thoroughly discussed. If they meet the same health standards and inclusion criteria as the males, I am unsure why females were excluded. Also, it appears that sex and gender are used interchangeably by using the term "males" and then "women"; please revise for consistency.
- Do you have the physical activity levels of the males included? If so, please add as it would provide valuable information in light of the denervation data.
- Line 114: It would improve clarity and understanding for the reader if it was explained why an enlargement of the nerve terminal area is considered a negative outcome. What are the repercussions of area growth?
- Line 124: It would be helpful if BeeNMJ was better explained on its first use.
- In the dietary section (line 175) of the methods an "ambulatory period" is mentioned, but this was never mentioned anywhere else, including the study design. What is this period? Was this after the BR10 timepoint? How long did it last? Was there a retraining period where they were brought back to baseline for ethical reasons?
- One part of the methods (line 171) made it seem like they weren't allowed to get up or even perform any voluntary movements, yet later it was stated that there was bathroom video monitoring. As such, it is unclear if they were allowed to get up and walk to use the bathroom or not. Please clarify, and change language accordingly regarding how limited their movements were.
- Line 201: how was the central motor point determined? Neither at this point, nor later when percutaneous electrical stimulation is mentioned (line 269) does it mention what type of stimulator was used. Please add
- Line 242: There were multiple images that were analyzed within ImageJ. Do the authors have reliability numbers for this analysis, which can be subjective?

- Line 328: In the results it is mentioned for the first time that there is another group, an intervention group, and some of their data is included in the results. The fact that this is part of a larger study needs to be mentioned much earlier (experimental design).
- Figure 2: Why were stats only performed on change scores? Was the change itself significant? This could have been provided had it been a 2x2 group by time RM ANOVA rather than a simpler model on change scores. Then we'd know if there was a main effect for time.
- Figure 3B: This is a fascinating analysis showing the distribution of Innervated, Partially Denervated, and Denervated NMJs. While this adds so much to this paper, it is almost disappointing that the first time I see this in older adult humans is in such a small sample size. This would be a hugely impactful analysis to do cross-sectionally across a large group of older adults (breaking them up into age intervals, such as 60-64, 65-69, 70-74, etc.), while also accounting for physical activity levels. Such a study would provide so much useful information regarding motor unit and NMJ remodeling in older adults. Please consider recommending such future work in your discussion section, because it could be extremely valuable to those in the field. Great job on this analysis.
- Line 399: should say "bed" rest
- I have concerns regarding all of the correlations performed. With such a small sample (6 data points), running over 40 correlations without any apparent alpha correction for multiple comparisons is concerning. By chance alone there should be a couple false positives (significant correlations). Furthermore, there are multiple non-significant correlations that are getting the attention of their own plot. Honestly, I do not believe these correlations add anything to this paper, and they run a high risk of being over-valued or misinterpreted. This paper is strong enough, in my opinion, without these low-sample correlations muddying up the results. That type of analysis should be save for larger-sample follow-up studies. As a result, I think line 590 stating that this data "supports the view that NMJ morphology and function are closely connected" may be stretch, or over-interpretation. I agree that they are likely closely connected, but using words such as "may be" should be considered to soften the boldness of this statement.

Referee #2:

Montanova and colleagues present a very interesting manuscript characterising neuromuscular remodelling as a consequence of 10 days bed rest in older male participants. The team utilise morphological, biochemical and functional approaches to demonstrate that the NMJ is functionally impaired after 10 days of inactivity. The findings are novel as NMJ remodelling in the context of inactivity in older individuals has not been extensively studied. Furthermore, the work may have future clinical impacts when considering the periods of time patients remain inactive following medical interventions or the sedentary lifestyle common amongst the elderly. The work contributes to enhancing the field's understanding of the physiological mechanisms of neuromuscular dysfunction associated with disuse. Experimental design, data collection, analysis and presentation appear appropriate. Conclusions seem valid but are based on data that currently have some methodological concerns (see below).

It would have been fascinating learn if the NMJs reverted back to the day 0 phenotype after a recovery period from 10 days bed rest. However, this is outside the scope of the current manuscript.

This reviewer has some concerns around the methodological details (see below), if these can be addressed appropriately, this manuscript will be an important addition to the literature.

Major concerns

1. Methods - despite the authors recognising the limitations of low n numbers, sampling of NMJs for morphological analysis must be addressed. A major limitation of many human NMJ studies is the low number of NMJs analysed in comparison to the number of fibres present in a muscle. How can the authors be sure that assessing an average of no more than 30 NMJs per biopsy is representative of the entire fibre population of the vastus lateralis muscle? This reviewer appreciates the technical difficulties and ethical implications in collecting large biopsies but some explanation supported by evidence is required to ensure these data are meaningful.

2. Methods - The authors have not accounted for myofibre type in these samples. How are the authors sure their analyses are not biased by fibre type composition in the biopsies taken? This is important as evidence suggests that NMJ denervation occurs in fast fibres compared to slow (<https://doi.org/10.1113/JP285143>). Furthermore, it will enhance the story if the authors analyse slow fibre type grouping which is often a marker of NMJ remodelling in muscle. In addition NCAM analysis could be conducted on muscle sections. The methods state that only a third of the biopsy was used for NMJ morphology. Was or could the remaining tissue be used for muscle fibre histology? This would also allow the authors to confirm muscle fibre atrophy in the samples strengthening the suggestion that lower motor unit size is a result of fibre atrophy (line 572).

3. Why did the authors not use an additional marker of the motor axon, such as neurofilament, for the morphological analysis? This is very standard practice when conducting NMJ morphological and has been previously reported in human tissue in the papers cited by the authors. This could have an impact on the analyses in Figure 4. The image shown in Figure 4F demonstrates an NMJ that appears to have expanded and is in line with the data presented in Figure 4A and B. However, how are the authors sure that the AchR cluster presented is indeed one NMJ? Including a marker such as neurofilament would allow clear visualisation of the motor axon branch colocalising (or not) with the AchR clusters demonstrating how many NMJs are present in a given field of view. Currently, it is difficult to interpret these images. Short of repeating the immuno, can the authors provide (in supplementary information) a collection of NMJ images analysed to address this point?

Minor comments

1. Throughout the manuscript the participants are referred to as 'individuals', given that females were not analysed, it is more accurate to describe all participants as 'male individuals'. Please correct throughout the manuscript.

2. Line 167, double full stop

3. Line 167, *in vivo* should be in italics

4. Methods - it is not stated how patients excreted waste over the 10-day period? Presumably, they did not walk to the bathroom? Please clarify as a considerable confounding factor would be if participants did not remain totally unloaded for the full 10 days.

5. Methods - there is no mention of blinding or randomisation protocols used when undertaking experiments or analysing data, can the authors please confirm these details?

6. The authors are to be commended on the level of transparency regarding the additional Old+interventions participants. They have shown no major effects on their experimental parameters (Fig2.) however, given the AKT/mTOR pathway is a known regulator of the NMJ and given this pathway is sensitive to amino acid intake, more details on the protein intervention

is recommended. Including the amount and frequency of protein given to participants would be sufficient for the readers to make their own conclusions.

7. Results - The text relating to figures 8, 9 and 10 are lacking description. The authors do not provide insights into what these data mean. This reviewer appreciates that most of the findings are non-significant but that still requires description as is given for results text relating to previous figures in the manuscript.

8. Discussion - line 486, the authors refer to the BR10 model as 'chronic' inactivity. This reviewer would consider chronic inactivity to occur over longer periods of time e.g. months or greater such as that associated with time in ICU or following major surgery etc. 10 days of inactivity seems more like an acute paradigm. Can the authors please explain their rationale for this description of 'chronic'?

END OF COMMENTS

EDITOR COMMENTS

Reviewing Editor:

This is a comprehensive study examining the effects of physical inactivity on neuromuscular junction structure and function in older individuals. It has been reviewed by two experts in neuromuscular physiology, particularly in the context of some of the experimental approaches you used (i.e., EMG and neuromuscular junction morphology assessed through biopsies). Both reviewers were generally favourable of your submission and have commented of the novelty of the work. Regardless, they raised some issues, and have made several constructive comments that need to be addressed. These issues primarily relate to the low sample size, and the fact that only male subjects were recruited. This affects how generalisable the findings are and need to be discussed in more detail to caution the over extrapolation of findings. There were also some issues raised regarding the methodology and statistical approach, which are likely addressed by re-analysing data/samples. I hope this feedback serves useful. As an additional observation, please note that error bars are not defined in Figure 2.

Please also see 'Required Items' above.

We are grateful to the Reviewing Editor for the expert handling of our manuscript and for identifying some areas for improvement together with the Reviewers. We have taken into consideration the Reviewers' suggestions as best as we could, in order to amend and improve the manuscript accordingly.

Senior Editor:

As noted by the referees and the Reviewing Editor, this work has many strengths. This being the case, it is recommended that the authors omit from their presentation, analyses that (due to the small sample size) are unlikely to appear convincing to readers. For example, and as highlighted in the reviews, the correlations have very low statistical power and, given the large number undertaken, are very likely to have yielded spurious false positive outcomes. More generally, the limitations on interpretation imposed by the small sample size should be emphasised.

We are grateful to the Senior Editor for coordinating the review process.

Please find our point-by-point response. Our comments/feedback are highlighted in grey. The corrections applied to the new version of the manuscript will be also highlighted in grey.

REFEREE COMMENTS

Referee #1:

This study examines NMJ morphology and function after 10 days of bed rest in older adult males. The authors are to be commended for the ambitious nature of this study. While the sample size is a bit low, the study design itself is very ambitious and provides incredibly

valuable information regarding changes to our NMJs during disuse. My comments and concerns are listed below:

We thank the Reviewer for taking the time to review our manuscript and for providing many valuable comments. We are pleased to address all the raised questions and concerns. We also appreciate the Reviewer's positive feedback and recognition of the value of our study.

- I do not understand why only males are included in this study. The only mention is at the very end, which it is stated that older women have a higher risk of frailty. I do think this needs to be justified and more thoroughly discussed. If they meet the same health standards and inclusion criteria as the males, I am unsure why females were excluded. Also, it appears that sex and gender are used interchangeably by using the term "males" and then "women"; please revise for consistency.

We thank the Reviewer for raising this important point. The decision was based on ethical considerations. Specifically, while both males and females who meet the same health standards and inclusion criteria could theoretically participate, older females face greater risks during muscle disuse studies, in particular during bed rest. This is due to their greater likelihood of adverse outcomes, such as accelerated bone loss, muscle atrophy, and increased frailty, particularly in post-menopausal older females. These risks make it ethically challenging to include older females, as current guidelines prioritise minimising harm to vulnerable populations. Additionally, females have an increased risk of deep vein thrombosis following long-term disuse interventions (<https://doi.org/10.1016/j.ijnurstu.2020.103825>). Although our intervention was only 10 days, obtaining ethical approval for older females to undergo even short-term inactivity periods remains challenging. We expanded the discussion on this aspect in the "Limitations" section (Lines 614-622). We would like to highlight that we are in the process of planning a new bed rest study, which will include both females and males from a younger population. As mentioned previously, due to ethical constraints regarding older females, this future study will focus solely on young adults to ensure participant safety and regulatory compliance.

We recognise the need to distinguish between sex and gender in our terminology. We revised the manuscript for consistency, using "males" and "females" to describe biological sex throughout the text to avoid confusion.

- Do you have the physical activity levels of the males included? If so, please add as it would provide valuable information in light of the denervation data.

Unfortunately, while we agree that having the physical activity levels would be beneficial for this study, they were not assessed at baseline. However, it is worth noting that the inclusion criteria of the study aimed at recruiting older non-sedentary individuals without any mobility impairments.

- Line 114: It would improve clarity and understanding for the reader if it was explained why an enlargement of the nerve terminal area is considered a negative outcome. What are the repercussions of area growth?

We thank the Reviewer for the suggestion. We have expanded this part of the introduction to clarify the role of an AChR area enlargement (Lines 97-111).

- Line 124: It would be helpful if BeeNMJ was better explained on its first use.

We have incorporated a brief description of the BeeNMJ technique on Lines 119-124. We agree that it is relevant to describe it better, as our biopsy sampling was inspired by the method developed by (Aubertin-Leheudre *et al.*, 2020).

- In the dietary section (line 175) of the methods an "ambulatory period" is mentioned, but this was never mentioned anywhere else, including the study design. What is this period? Was this after the BR10 timepoint? How long did it last? Was there a retraining period where they were brought back to baseline for ethical reasons?

We thank the Reviewer for allowing us to clarify this point. The "ambulatory period" refers to the baseline measurement period, during which participants came to the hospital 2 days before the bed rest onset to familiarise with the procedures and perform the initial *in vivo* assessments. We adjusted this information in the text on Lines 181-182. Following the bed rest period, participants returned home but were required, for ethical reasons, to complete a 3-week aerobic retraining program at local gyms to help restore baseline fitness levels.

- One part of the methods (line 171) made it seem like they weren't allowed to get up or even perform any voluntary movements, yet later it was stated that there was bathroom video monitoring. As such, it is unclear if they were allowed to get up and walk to use the bathroom or not. Please clarify, and change language accordingly regarding how limited their movements were.

We thank the Reviewer for the comment. We have clarified these aspects by adjusting the text in Lines 177-180. During the bed rest period, participants remained in a horizontal position at all times, including during personal hygiene and toileting activities. Bathroom video monitoring was used solely to ensure participants did not stand or walk. Participants used a horizontal shower, and for toileting activities, they used specialised containers.

- Line 201: how was the central motor point determined? Neither at this point, nor later when percutaneous electrical stimulation is mentioned (line 269) does it mention what type of stimulator was used. Please add

We thank the Reviewer for allowing us to clarify this point. Stimulations were delivered using a pen electrode connected to a Digitimer DS7AH stimulator device, applying an electrical current of 16 mA (400 V) with a pulse width of 100 μ s. After localising the motor point, the current was lowered to 8–10 mA to confirm it as the most responsive site, producing the largest twitch. We added this information at Lines 281-284.

- Line 242: There were multiple images that were analyzed within ImageJ. Do the authors have reliability numbers for this analysis, which can be subjective?

We thank the Reviewer for the valuable comment. The main parameter of NMJ morphology considered in this study was the overlap between the presynaptic and postsynaptic terminals, which is a critical measure of NMJ integrity. This analysis was conducted using an automated

plugin in ImageJ, which detects the signal independently, making it an objective and reliable measurement free from operator bias.

For the AChR area and perimeter, which were assessed manually, we do not currently have reliability measurements. However, we took steps to minimise potential bias. The analysis was conducted double-blindly, with the IDs of NMJ images renamed by an independent operator before analysis, ensuring the primary operator was blind to the specific condition. This process helped further reduce subjectivity in these manual measurements.

- Line 328: In the results it is mentioned for the first time that there is another group, an intervention group, and some of their data is included in the results. The fact that this is part of a larger study needs to be mentioned much earlier (experimental design).

We thank the Reviewer for the comment. We added the information on the intervention group in the “Participants” section (Lines 151-159).

- Figure 2: Why were stats only performed on change scores? Was the change itself significant? This could have been provided had it been a 2x2 group by time RM ANOVA rather than a simpler model on change scores. Then we'd know if there was a main effect for time.

We thank the Reviewer for the valuable suggestion. We have replaced the previous graph (Fig.2) with a new one based on a 2x2 group*time repeated measure ANOVA analysis. Initially, we used fold changes to simplify the graphic representation, but we agree that the new analysis ensures better transparency regarding the absence of differences between the two groups over time.

- Figure 3B: This is a fascinating analysis showing the distribution of Innervated, Partially Denervated, and Denervated NMJs. While this adds so much to this paper, it is almost disappointing that the first time I see this in older adult humans is in such a small sample size. This would be a hugely impactful analysis to do cross-sectionally across a large group of older adults (breaking them up into age intervals, such as 60-64, 65-69, 70-74, etc.), while also accounting for physical activity levels. Such a study would provide so much useful information regarding motor unit and NMJ remodeling in older adults. Please consider recommending such future work in your discussion section, because it could be extremely valuable to those in the field. Great job on this analysis.

We thank the Reviewer for the enthusiasm regarding this analysis and the insightful comment. We fully agree that the sample size is quite small, but given the technical difficulties in identifying NMJs in human biopsies, even with the motor point technique, it is indeed challenging to find positive samples. We were fortunate to identify six participants with matching pre- and post-measurements, and we imaged all the NMJs available within the biopsies.

We also agree that a cross-sectional study would be extremely valuable, not just in older adults but also in younger and middle-aged individuals. While current evidence suggests that NMJs remain stable across the lifespan, they seem to be affected by inactivity, making it crucial to account for physical activity levels, so it would be interesting to see if that is the case also in other age groups.

We have incorporated this suggestion into our discussion section (Lines 507-516) and thank the Reviewer for this excellent recommendation, which we fully support.

- Line 399: should say "bed" rest

We thank the Reviewer for spotting this typo. We have corrected it.

- I have concerns regarding all of the correlations performed. With such a small sample (6 data points), running over 40 correlations without any apparent alpha correction for multiple comparisons is concerning. By chance alone there should be a couple false positives (significant correlations). Furthermore, there are multiple non-significant correlations that are getting the attention of their own plot. Honestly, I do not believe these correlations add anything to this paper, and they run a high risk of being over-valued or misinterpreted. This paper is strong enough, in my opinion, without these low-sample correlations muddying up the results. That type of analysis should be save for larger-sample follow-up studies. As a result, I think line 590 stating that this data "supports the view that NMJ morphology and function are closely connected" may be stretch, or over-interpretation. I agree that they are likely closely connected, but using words such as "may be" should be considered to soften the boldness of this statement.

We thank the Reviewer for their valuable suggestion. To minimise emphasis on these results, we removed the figures depicting the observed correlations and omitted their mention in the abstract. We have also highlighted in both the results and discussion sections that the results of the correlation analysis should be interpreted with caution (Lines 456-461; 469-477; 588; 598; 600-602).

Referee #2:

Montanova and colleagues present a very interesting manuscript characterising neuromuscular remodelling as a consequence of 10 days bed rest in older male participants. The team utilise morphological, biochemical and functional approaches to demonstrate that the NMJ is functionally impaired after 10 days of inactivity. The findings are novel as NMJ remodelling in the context of inactivity in older individuals has not been extensively studied. Furthermore, the work may have future clinical impacts when considering the periods of time patients remain inactive following medical interventions or the sedentary lifestyle common amongst the elderly. The work contributes to enhancing the field's understanding of the physiological mechanisms of neuromuscular dysfunction associated with disuse. Experimental design, data collection, analysis and presentation appear appropriate. Conclusions seem valid but are based on data that currently have some methodological concerns (see below).

It would have been fascinating learn if the NMJs reverted back to the day 0 phenotype after a recovery period from 10 days bed rest. However, this is outside the scope of the current manuscript.

This reviewer has some concerns around the methodological details (see below), if these can be addressed appropriately, this manuscript will be an important addition to the literature.

We thank the Reviewer for taking the time to review our manuscript and for the positive feedback on our study. We have tried to take into consideration the Reviewers' suggestions as best as we could.

Major concerns

1. Methods - despite the authors recognising the limitations of low n numbers, sampling of NMJs for morphological analysis must be addressed. A major limitation of many human NMJ studies is the low number of NMJs analysed in comparison to the number of fibres present in a muscle. How can the authors be sure that assessing an average of no more than 30 NMJs per biopsy is representative of the entire fibre population of the vastus lateralis muscle? This reviewer appreciates the technical difficulties and ethical implications in collecting large biopsies but some explanation supported by evidence is required to ensure these data are meaningful.

We thank the Reviewer for their thoughtful comments on the representativeness of our NMJ sampling strategy. We agree that the number of NMJs analysed is a critical point in assessing the overall integrity of the NMJs across the entire muscle. However, we believe that our sample size and method of sampling are supported by several previous important studies in the field, which we will explain in further detail.

Our sampling of approximately 30 NMJs per participant is consistent with several well-established studies in human NMJ research. For example, Jones et al. (2017; <https://doi.org/10.1016/j.celrep.2017.11.008>.) analysed approximately 40 NMJs per muscle in an ageing study, and Boehm et al. (2020; <https://doi.org/10.1172/JCI128411>.) utilised a similar number of NMJs per participant in a study on cancer cachexia. Additionally, Slater et al. (1992; PMID: 1351415) used the motor point biopsy technique and analysed 10 NMJs per individual in a study on NMJ structure in the vastus lateralis. Furthermore, Aubertin-Leheudre et al. (2020; <https://doi.org/10.1093/gerona/glz292>.) also observed an average of 30 NMJs per sample in their investigation. Overall, these studies provide evidence that examining 30–40 NMJs is sufficient for drawing meaningful conclusions about NMJ morphology and stability in humans.

As acknowledged in the manuscript, obtaining larger biopsies from human subjects (especially elderly people) is subjected to ethical limitations and presents technical challenges. Unlike animal models, where whole muscles can be dissected, human biopsies are limited in size, particularly when using the motor point biopsy technique, which restricts the number of NMJs that can be analysed. While animal models allow for more extensive sampling due to the ability to access longer, undamaged muscle fibres, the human muscle biopsy method requires a more constrained approach.

Moreover, studies examining the vastus lateralis muscle have shown that fibre type distribution is random across different fascicles. For instance, Lexell et al. (1983; [https://doi.org/10.1016/0022-510X\(83\)90247-9](https://doi.org/10.1016/0022-510X(83)90247-9).) demonstrated that fibre type arrangement within the vastus lateralis is random, with no significant differences in fibre type proportions between various regions of the muscle. This suggests that sampling NMJs from a single motor point, as we did, is likely representative of the entire muscle. The random distribution of fibre

types supports the notion that NMJ structure and function from one region can reliably reflect the broader muscle architecture. Thus, we believe our sampling strategy is consistent with findings from multiple studies showing that analysing 30–40 NMJs provides a meaningful and representative assessment of NMJ integrity across the muscle. Additionally, we were always acquiring our samples from the same region of the muscle, the participants before and after bedrest coincided.

2. Methods - The authors have not accounted for myofibre type in these samples. How are the authors sure their analyses are not biased by fibre type composition in the biopsies taken? This is important as evidence suggests that NMJ denervation occurs in fast fibres compared to slow (<https://doi.org/10.1113/JP285143>). Furthermore, it will enhance the story if the authors analyse slow fibre type grouping which is often a marker of NMJ remodelling in muscle. In addition NCAM analysis could be conducted on muscle sections. The methods state that only a third of the biopsy was used for NMJ morphology. Was or could the remaining tissue be used for muscle fibre histology? This would also allow the authors to confirm muscle fibre atrophy in the samples strengthening the suggestion that lower motor unit size is a result of fibre atrophy (line 572).

We appreciate the Reviewer's thoughtful comments and suggestions regarding our study. Below, we provide further clarification on the several points raised:

We acknowledge the importance of considering myofibre type when analysing NMJs, particularly in light of the differential susceptibility of fast- and slow-twitch fibres to NMJ denervation. However, in our study, we were unable to perform fibre typing analysis. This is because the biopsy sample was dedicated to different analyses and techniques performed by other research groups involved in the bed rest campaign, not solely for the NMJ analysis performed by us. Unfortunately, the method we use for staining and categorising NMJs is not compatible with myosin heavy chain staining, as it requires teasing the muscle sample into small fibre bundles rather than cutting and staining cross-sections from OCT-embedded pieces, which is possible in rodent muscles. Indeed, human samples are not as enriched with NMJs as rodent samples, making it almost impossible to detect NMJs in cross-sectional slices, and fibre typing typically requires cross-sectional muscle samples. Our focus was to acquire as many NMJs as possible, and sacrificing a piece of tissue for other analyses would have resulted in an even smaller mean NMJ number per participant, which you have reasonably identified as a limitation in your previous comment. Moreover, if the muscle is mixed, such as the vastus lateralis, containing both slow- and fast-twitch fibres, our NMJ staining technique, which requires whole muscle fibres, would still not allow us to distinguish between NMJs innervating either slow or fast fibres, as fibre typing requires cross-sections, which would have been done as a separate analysis. Furthermore, unlike in rodents, human vastus lateralis muscle fibres are more or less equally distributed, meaning that the biopsy would contain both fibre types. We would also like to emphasise once again that the biopsy samples were acquired from the same small motor point region of the muscle in participants before and after bed rest.

Regarding NCAM staining, while we have also widely used it and consider it an interesting biomarker, we believe it would not add much in the context of the present study. Indeed,

the use of NCAM to evaluate NMJ innervation status has been questioned due to its indirect nature (<https://doi.org/10.1002/jcsm.13624>). As our manuscript focuses on direct visualisation and a more detailed categorisation of NMJs using an automated plugin, we believe this approach provides a direct assessment of NMJ changes, making the addition of NCAM unlikely to provide substantial additional value. Moreover, we have already reported NCAM changes with 10 days of disuse across bed rest (<https://doi.org/10.1113/JP281365>) and unilateral lower limb suspension (<https://doi.org/10.1113/JP283381>) studies; thus, we believe its addition would provide limited additional novelty.

We agree that the study would indeed benefit from fibre type grouping analysis, and, unfortunately, we do not have cross-sections of OCT-embedded muscle samples available for this analysis. This limitation, as mentioned earlier, was due to the specific focus of the study. We acknowledge that fibre type analysis could have added another layer to our investigation, and we regret not being able to perform it. However, in our previous study, we had a chance to evaluate it in another disuse model in young individuals, and both slow and fast fibre grouping were unchanged following a 10-day unloading period (<https://doi.org/10.1113/JP283800>). Thus, we believe it is unlikely that fibre type grouping could have occurred within such a short disuse period in the current investigation.

Regarding muscle fiber atrophy, we relied on other studies showing that fibre atrophy occurs after more than one week of bed rest in older adults (<https://doi.org/10.1093/gerona/glaa001>; doi: 10.1002/jcsm.12306; doi: 10.2337/db15-1661).

3. Why did the authors not use an additional marker of the motor axon, such as neurofilament, for the morphological analysis? This is very standard practice when conducting NMJ morphological and has been previously reported in human tissue in the papers cited by the authors. This could have an impact on the analyses in Figure 4. The image shown in Figure 4F demonstrates an NMJ that appears to have expanded and is in line with the data presented in Figure 4A and B. However, how are the authors sure that the AchR cluster presented is indeed one NMJ? Including a marker such as neurofilament would allow clear visualisation of the motor axon branch colocalising (or not) with the AchR clusters demonstrating how many NMJs are present in a given field of view. Currently, it is difficult to interpret these images. Short of repeating the immuno, can the authors provide (in supplementary information) a collection of NMJ images analysed to address this point?

Thank you for your insightful comments and for raising these important points.

To be fully transparent, we want to clarify that we did indeed perform staining for axons using neurofilament, and this has now been included in the Methods section of our manuscript (Lines 220,225,244). However, we did not utilise this staining for quantification. This decision was based on concerns specific to human NMJs obtained from biopsies, where the axons often appear distorted. We were thus cautious about the reliability of manual axon thickness analysis under such conditions. Additionally, studies in mice have demonstrated that, in some cases, NMJs display presynaptic vesicle staining even when the axon has already retracted

(<https://doi.org/10.7554/eLife.41973>). This further informed our decision to exclude this parameter from the primary analysis.

That said, neurofilament staining was indeed employed to help distinguish between individual NMJs, as suggested. However, it is also possible to distinguish NMJs without neurofilament staining. Since each NMJ innervates a single muscle fibre (<https://doi.org/10.1016/j.ncl.2018.01.009>), we were also able to reliably differentiate between junctions using alpha-bungarotoxin labelling of AChR.

For your convenience, we have uploaded a supplementary figure below that includes neurofilament staining to further illustrate the NMJs, as requested. These NMJs are the same ones provided in Fig.4 in the main manuscript. Additionally, we are openly sharing the datasets of all our z-stacks for further transparency in The Open Science Framework (OSF). The link to this database is included in the Data Availability section of our manuscript.

We hope this addresses your concerns and enhances the clarity of our approach.

Minor comments

1. Throughout the manuscript the participants are referred to as 'individuals', given that females were not analysed, it is more accurate to describe all participants as 'male individuals'. Please correct throughout the manuscript.
2. Line 167, double full stop
3. Line 167, *in vivo* should be in italics

We thank the Reviewer for pointing this out. We have amended the manuscript based on minor comments 1, 2 and 3 accordingly.

4. Methods - it is not stated how patients excreted waste over the 10-day period? Presumably, they did not walk to the bathroom? Please clarify as a considerable confounding factor would be if participants did not remain totally unloaded for the full 10 days.

We thank the Reviewer for the comment. We have clarified this information in the manuscript to address this concern (Lines 177-180). During the bed rest period, participants remained in a horizontal position at all times, including during personal hygiene and toileting activities. Bathroom video monitoring was conducted to ensure participants did not stand or walk. A male urinal and fracture bedpans were used for urination and defecation, respectively, both done in a supine position in a separate room – a washing room/toilet. Also showering was done in a supine position in a horizontal shower.

5. Methods - there is no mention of blinding or randomisation protocols used when undertaking experiments or analysing data, can the authors please confirm these details?

The analysis of NMJ morphology was conducted double-blindly, with NMJ images IDs renamed by an independent operator before analysis, ensuring the primary operator was unaware of specific labelling to perform an unbiased assessment of the images. We have included this information in the manuscript (Lines 239-241). For iEMG analysis, blinding was not applied to the operator.

6. The authors are to be commended on the level of transparency regarding the additional Old+interventions participants. They have shown no major effects on their experimental parameters (Fig2.) however, given the AKT/mTOR pathway is a known regulator of the NMJ and given this pathway is sensitive to amino acid intake, more details on the protein intervention is recommended. Including the amount and frequency of protein given to participants would be sufficient for the readers to make their own conclusions.

Please find the details regarding the protein intake in the attachment, and we have also included this information in the manuscript (Lines 156-159): during the bed rest period, the intervention group followed a high-protein diet enriched with BCAAs. Dietary protein intake was increased to approximately $1.45 \text{ g}\cdot\text{kg}^{-1}\cdot\text{day}^{-1}$ using a high-protein oral nutritional supplement (Fortifit, Nutricia, Italy), enriched with $0.15 \text{ g}\cdot\text{kg}^{-1}\cdot\text{day}^{-1}$ of BCAAs in a 2:1:1 ratio (leucine/valine/isoleucine). Additional free BCAAs were provided in the form of supplements (Friliver, Bracco, Italy) with doses of 3.6 g/day leucine, 1.8 g/day isoleucine, and 1.8 g/day valine, administered during the three main meals.

As shown in the new Fig. 2, the overlap between the two NMJ terminals was analysed to assess the effects of the disuse intervention between the two groups. In both groups, the overlap between NMJ terminals was reduced. However, there were no significant differences in the time*group interaction, indicating that the supplementation had no visible effect on the NMJs morphology in the old+int group. Thus, the supplementation did not appear to have a protective effect on NMJ integrity. This was additionally confirmed by the similar behaviour in CAF analysis.

7. Results - The text relating to figures 8, 9 and 10 are lacking description. The authors do not provide insights into what these data mean. This reviewer appreciates that most of the

findings are non-significant but that still requires description as is given for results text relating to previous figures in the manuscript.

We thank the Reviewer for the valuable suggestion. To address the comments from Reviewer 1, we have decided to remove the images of the observed correlations to reduce the focus on these results, as they must be interpreted with extreme caution due to the small sample size. We have also omitted the mention of correlations in the abstract. Instead, we have expanded the explanation of these results in the results and discussion sections (Lines 456-461; 469-477; 588; 598; 600-602).

8. Discussion - line 486, the authors refer to the BR10 model as 'chronic' inactivity. This reviewer would consider chronic inactivity to occur over longer periods of time e.g. months or greater such as that associated with time in ICU or following major surgery etc. 10 days of inactivity seems more like an acute paradigm. Can the authors please explain their rationale for this description of 'chronic'?

In physical inactivity research, 'acute' interventions typically refer to inactivity protocols lasting less than one day, such as prolonged sitting (<https://doi.org/10.1152/physrev.00022.2022>). Based on this concept, our 10-day bed rest study could be considered a chronic intervention. However, for clarity, we have removed the term "chronic" throughout the manuscript.

Dear Miss Motanova,

Re: JP-RP-2025-288448R1 "Neuromuscular Junction Instability with Inactivity: Morphological and Functional Changes After 10-Day Bed Rest in Older Adults" by Evgeniia Motanova, Fabio Sarto, Samuele Negro, Marco Pirazzini, Ornella Rossetto, Michela Rigoni, Dan Stashuk, Mladen Gasparini, Bostjan Simunic, Rado Pisot, and Marco Narici

We are pleased to tell you that your paper has been accepted for publication in The Journal of Physiology.

Yours sincerely,

Richard Carson
Senior Editor
The Journal of Physiology

If you would like to receive our 'Research Roundup', a monthly newsletter highlighting the cutting-edge research published in The Physiological Society's family of journals (The Journal of Physiology, Experimental Physiology, Physiological Reports, The Journal of Nutritional Physiology and The Journal of Precision Medicine: Health and Disease), please click this link, fill in your name and email address and select 'Research Roundup':
<https://www.physoc.org/journals-and-media/membernews>

- You can help your research get the attention it deserves! Check out Wiley's free Promotion Guide for best-practice recommendations for promoting your work at: www.wileyauthors.com/eeo/guide. You can learn more about Wiley Editing Services which offers professional video, design, and writing services to create shareable video abstracts, infographics, conference posters, lay summaries, and research news stories for your research at: www.wileyauthors.com/eeo/promotion.

EDITOR COMMENTS

Reviewing Editor:

The authors have addressed all concerns and questions raised by the expert reviewers. I commend the authors on their work.

REFEREE COMMENTS

Referee #1:

I am satisfied with the revisions, and all of my concerns have been adequately addressed. I have no further comments or concerns. Great work.

Referee #2:

The authors have adequately addressed all my comments.